# Derivation and validation of a refined dust product from Aeolus (L2A+)

Konstantinos Rizos[1], Emmanouil Proestakis[1], Thanasis Georgiou[1], Antonis Gkikas[1,2], Eleni Marinou[1], Peristera Paschou[1,3], Kalliopi Artemis Voudouri[1,3], Athanasios Tsikerdekis[4], David P. Donovan[4], Gerd-Jan van Zadelhoff[4], Angela Benedetti[5], Holger Baars[8], Athena Augusta Floutsi[8], Nikos Benas[4], Martin Stengel[6], Christian Retscher[7], Edward Malina[7], Vassilis Amiridis[1]

1 Institute for Astronomy, Astrophysics, Space Applications and Remote Sensing, National Observatory of Athens, 15236 Athens, Greece
2 Research Centre for Atmospheric Physics and Climatology, Academy of Athens, Athens, 10680, Greece
3 Laboratory of Atmospheric Physics, Aristotle University of Thessaloniki, 54124 Thessaloniki, Greece
4 Royal Netherlands Meteorological Institute (KNMI), de Bilt, the Netherlands
5 European Centre for Medium-Range Weather Forecasts (ECMWF), Shinfield Park, Reading RG2 9AX, UK
6 Deutscher Wetterdienst (DWD), Offenbach, Germany
7 European Space Agency (ESA/ESRIN), 00044 Frascati, Italy
8 Leibniz Institute for Tropospheric Research (TROPOS), Leipzig, Germany

Correspondence to: Konstantinos Rizos (k.rizos@noa.gr)

**Abstract.** Originally designed as a wind lidar without depolarization measurement capability, ALADIN (Atmospheric Laser Doppler Instrument) detects only the co-polar component of the backscattered signal, limiting the accuracy of optical products in the presence of depolarizing atmospheric layers. The absence of the cross-polar component also prohibits ALADIN's ability to distinguish between different aerosol and cloud types, in its retrievals. To address these limitations, an enhanced Aeolus aerosol product (L2A+), with a focus on dust, has been developed in the present study to support aerosol data assimilation in dust transport models and improve Numerical Weather Prediction (NWP). The L2A+ product is generated through a series of processing steps that integrate multi-sensor satellite retrievals for cloud screening and aerosol layer characterization, CAMS reanalysis outputs to classify aerosol types, distinguish dust from non-dust fractions, and provide the missing depolarization ratio values required for the Polarization Lidar Photometer Networking (POLIPHON) technique, along with ground-based lidar measurements used for performance assessment.Both the primary (L2A) and enhanced (L2A+) Aeolus pure-dust backscatter coefficient profiles at 355 nm are retrieved using four different algorithms: the Standard Correct Algorithm (SCA), the Standard Correct Algorithm with middle-bin vertical scaling (SCA-MB), the Maximum Likelihood Estimation (MLE), and AEL-PRO.These products are validated against ground-based reference observations obtained from the eVe and Polly[XT] lidar systems, operated as part of the ASKOS/JATAC experimental campaign in Mindelo, Cabo Verde. The approach is detailed on the basis of an indicative Aeolus overpass in the proximity of Mindelo on September 3, 2021, discussing ALADIN's sources of underestimation in terms of L2A backscatter coefficient at 355 nm profiles in the presence of desert dust particles across all four retrieval algorithms and the induced improvements achieved by accounting for the missing cross-polar component. A statistical evaluation of all Aeolus overpasses during the entire ASKOS/JATAC campaign in the Cabo Verde/Mindelo region confirms the enhanced performance of the upgraded L2A+ product compared to the original L2A product. This improvement is evident in both Aeolus-eVe and Aeolus-Polly[XT] comparisons across all retrieval algorithms and is marked by higher correlation coefficients and regression slopes along with lower biases and RMSE scores. Specifically, among the algorithms, the AEL-PRO and MLE L2A+ products show significant improvements in alignment with eVe lidar observations. The correlation coefficients increased from 0.59 to 0.67 for MLE and from 0.55 to 0.67 for AEL-PRO, while biases decreased from -0.99 to -0.85 Mm⁻¹sr⁻¹ for MLE and from -0.79 to -0.58 Mm⁻¹sr⁻¹ for AEL-PRO. They also achieve lower RMSE values (1.17 Mm⁻¹sr⁻¹ for MLE and 0.94 Mm⁻¹sr⁻¹ for AEL-PRO) and better regression slopes, increasing from 0.28 to 0.34 (MLE) and 0.27 to 0.39 (AEL-PRO). Similarly, L2A+ adjustments reduce biases and improve correlation coefficients, and regression slopes in Aeolus-Polly[XT] comparisons, among all four retrieval algorithms. Particularly, the AEL-PRO algorithm shows the strongest improvement in correlation, slope, and error metrics. All these advancements establish the enhanced L2A+

dust product as a strong candidate for aerosol data assimilation, supporting improved dust transport modeling and further enhancing Numerical Weather Prediction (NWP).

## 1. Introduction


Atmospheric mineral dust, the second-most abundant aerosol type in the global atmosphere, plays a key role in climate and atmospheric chemistry. It directly affects the radiation balance of the planet through the scattering or absorption of the sunlight (Ghan et al., 2012; Haywood & Boucher, 2000; Myhre, 2009), and it also has an indirect effect through its interactions with clouds, acting as cloud condensation nuclei (CCN; Hatch et al., 2008) and/or ice-nucleating
particles (INPs; DeMott et al., 2010; Marinou et al., 2019), thereby modulating the reflectivity and lifetime of the cloud (Andreae & Crutzen, 1997; Lohmann & Feichter, 2005). Mineral dust has a beneficial role for marine and terrestrial ecosystems being a major source of essential nutrients like iron (Fe) and phosphorus (P) for oceans and land ecosystems upon deposition (Okin et al., 2004; Li et al., 2018). However, at high concentrations it can cause air quality degradation (Kanakidou et al., 2011; Proestakis et al., 2024), and it can also have detrimental consequences on the
human health with adverse respiratory, cardiovascular, cardiopulmonary and other severe diseases (Pope & Dockery, 2006; Contini et al., 2021; Korhonen et al., 2021).

It is estimated that each year approximately 4680Tg of mineral dust particles (Kok et al., 2023) are emitted from arid and semi-arid regions of the planet into the atmosphere (Washington et al., 2003; Schepanski et al., 2007; Yu et al., 2013; Varga et al., 2021). In particular, the dust emissions from Middle East and Asia account for ~ 12 % and 13 %
of the global emissions, respectively, while the Saharan desert is considered as the major contributor to the dust budget around the globe accounting for more than 50% of the global dust (Tanaka & Chiba, 2006; Ginoux et al., 2012; Lian et al., 2022), with its most intense source being the Bodélé Depression in the northern Lake Chad Basin (Gkikas et al., 2021). In North Africa, substantial quantities of mineral particles are also released from the western Sahara, while smaller but significant sources are found in the eastern Libyan Desert, the Nubian Desert (Egypt), and Sudan
(Engelstaedter et al., 2006). Once uplifted and inserted in the atmosphere, mineral dust particles can be transported on intercontinental, hemispherical, and even global scales driven by the prevailing winds (Goudie & Middleton, 2006; Marinou et al., 2017; Proestakis et al., 2018). On a seasonal basis, Saharan dust particles under the prevailing trade winds can travel across the Atlantic Ocean impacting the air quality in the Caribbean Basin, Central America, and the southern United States during boreal summer and South America during spring and winter (Prospero & Nees, 1986;
Karyampudi et al., 1999; Kalashnikova & Kahn, 2008; Gkikas et al., 2021, 2022; Mehra et al., 2023). Moreover, the Eastern Mediterranean is another region particularly vulnerable to Saharan dust transport. Studies have shown that mineral dust particles originating from the arid regions of North Africa and the Arabian Peninsula frequently impact air quality in the eastern and central Mediterranean, leading to elevated background pollution levels, especially during the spring and summer months (Gerasopoulos et al., 2006; Papayannis et al., 2008; Gkikas et al., 2015; Rizos et al.,
85 2022).

Given the key role of dust aerosols in the Earth system, global and routine measurements of dust extending over years or even decades are essential for examining dust emission, transport, and deposition processes, allowing estimations on dust radiative effects, as well as evaluating and constraining dust simulations in numerical weather and climate models (Song et al., 2021). Towards achieving to a certain degree this objective, satellites are utilized allowing for
accurate, comprehensive, real-time observations and global coverage over extended time periods. To date, passive satellite sensors, providing columnar retrievals of aerosol optical depth (AOD), have been used extensively to provide dust aerosol loads across various spatiotemporal scales (Kalashnikova & Kahn, 2008; H. Yu et al., 2009; Ginoux et al., 2010; Clarisse et al., 2019; Zhou et al., 2020). However, while these passive satellite sensors can provide global or quasi-global coverage of column-integrated aerosol optical properties with high temporal and spatial resolution, they
are not suitable for capturing information in the vertical scale that is critical for depicting the vertical structure of the transported dust loads within the planetary boundary layer or in the free troposphere (Gkikas et al., 2018, 2023; Q. Song et al., 2021). The aforementioned observational gaps of passive sensors are addressed by the utilization of observations acquired from space-borne lidar systems, such as the Cloud-Aerosol Lidar with Orthogonal Polarization (CALIOP) aboard the Cloud Aerosol Lidar and Infrared Pathfinder Satellite Observation (CALIPSO) satellite (Winker
et al., 2009), and the Cloud Aerosol Transport System (CATS) mounted on the International Space Station (ISS) (Lee et al., 2019; Pauly et al., 2019; Proestakis et al., 2019), able to provide the vertical structure of aerosol and clouds (Amiridis et al., 2013; Huang et al., 2015; Marinou et al., 2017; Proestakis et al., 2024).

The European Space Agency (ESA) Aeolus mission, which operated from 22 August 2018 to 28 July 2023, was designed to improve our understanding of wind patterns in the Earth's atmosphere. It carried the ALADIN (Atmospheric LAser Doppler INstrument) instrument, the first space-based and state-of-the-art High Spectral Resolution Lidar (HSRL) Doppler wind lidar operating at 355 nm wavelength. The primary mission goal was to provide vertically-resolved measurements of wind profiles in the troposphere and lower stratosphere at global scale, allowing to address open scientific questions (Martin et al., 2023), and to improve the numerical weather prediction (NWP) (Flament et al., 2021). A key advantage of the applied HSRL method is that it enables the global monitoring of aerosol and cloud optical properties through the independent estimation of the volume extinction coefficient and co-polarized volume backscatter coefficient optical products at 355 nm from two different spectral channels implementing robust crosstalk corrections to separate the molecular and particle signals (Ehlers et al., 2022; Flamant et al., 2008). In Aeolus, the retrieval of the atmospheric optical properties was implemented in the Level 2A (L2A) processor (Flament et al., 2021).

The L2A aerosol and cloud optical product retrievals from ALADIN have been systematically validated against a variety of independent reference ground-based measurements (Baars et al., 2021; Paschou et al., 2022; Abril-Gago et al., 2022; Gkikas et al., 2023). Baars et al. (2021) revealed an excellent agreement between the Aeolus' backscatter coefficient at 355 nm, extinction coefficient at 355 nm, and lidar ratio at 355 nm profiles retrieved from the Standard Correct Algorithm (SCA) and ground-based concurrent observations acquired from the Polly[XT] lidar (Engelmann et al., 2016) during a case of long-range transport of wildfire smoke particles from California (USA) to Leipzig (Germany). In a later attempt, Abril-Gago et al. (2022) carried out a validation of the Aeolus co-polar backscatter coefficients reprocessed with the L2A processor version 3.10, referred as Baseline 10 (B10),against reference ground-based measurements acquired from three EARLINET (European Aerosol Research Lidar Network; Pappalardo et al., 2010; last visit: 16/01/2025) monitoring stations mainly influenced by dust and continental/anthropogenic aerosols during the period between July 2019 and October 2020. The statistical analysis revealed the ability of the Aeolus lidar system to identify and characterize significant aerosol layers under cloud-free conditions with the Standard Correct Algorithm middle-bin (SCA-MB) presenting a better agreement with ground-based observations than the Standard Correct Algorithm (SCA). However, Aeolus L2A performance is reduced when depolarizing atmospheric features (i.e. dust particles, volcanic ash and cirrus ice crystals) are probed. More specifically, Paschou et al. (2022), on the basis of a dust intrusion event over Athens-Greece on September 24, 2020, reported underestimation of the order of 18% of the Aeolus-like backscatter coefficient at 355 nm on the atmospheric dust layer as provided by the ground-based Aeolus-reference ESA-eVe lidar system. In addition, Gkikas et al. (2023) reported, on the basis of EARLINET-Antikythera Polly[XT] observations of Saharan dust layer, similar performance of Aeolus SCA backscatter coefficient at 355 nm profiles, with an underestimation in the range from 13 % to 33 %. Recent validation results from Paschou et al. (2025)also revealed that within the 2.3–5.3 km altitude range, where dust particles predominantly reside during the ASKOS campaign (Marinou et al., 2023), all Aeolus retrieval algorithms (SCA, MLE, and AEL–PRO) exhibited an overall underestimation of the co-polar particle backscatter coefficient, with the most pronounced biases observed for the SCA algorithm and the smallest for MLE. The reported L2A backscatter coefficient underestimations were attributed to the missing cross-channel of the ALADIN, hampering the capacity to obtain realistic optical products when non-spherical particles (i.e. dust) were probed. Moreover, the absence of cross-polar component measurements on ALADIN's backscatter detected signals prohibits provision of particulate depolarization ratio profiles, limiting atmospheric feature-type and aerosol-subtype classification efforts (Song et al., 2023).

Towards overcoming the limitations of ALADIN lidar attributed to the missing cross-polar component, the present study delivers an upgraded Aeolus L2A aerosol product (L2A+ hereinafter) focusing on dust aerosol. For the development of this new Aeolus dust product, a multi-step approach has been implemented, combining spaceborne retrievals, CAMS reanalysis outputs, and reference ground-based observations. Specifically, Aeolus L2A backscatter profiles at 355 nm, retrieved with four algorithms—namely the Standard Correct Algorithm (SCA), the Standard Correct Algorithm middle-bin (SCA-MB), the Maximum Likelihood Estimation (MLE), and the AEL-PRO—have been processed to correct for the missing cross-polar contribution. CAMS data are used to provide aerosol-type classification and particle depolarization ratios required for the implementation of the one-step Polarization Lidar Photometer Networking (one-step POLIPHON; Ansmann et al., 2019; Tesche et al., 2009) method, developed within the framework of the European Aerosol Research Lidar Network (EARLINET; Pappalardo et al., 2010). This approach enables the decoupling of the atmospheric dust component from the total aerosol load, allowing for the estimation of the dust mass concentration even when dust does not represent the entire aerosol mixture in each layer (Ansmann et al., 2019). Ground-based lidar observations from the eVe and Polly[XT] lidars are also employed in the present study to evaluate and validate the L2A+ dust product, providing an assessment of its accuracy. The study period refers to the

period when the ASKOS experiment of the Joint Aeolus Tropical Atlantic Campaign (JATAC) was implemented in the Cabo Verde islands during the summer/autumn of 2021 and 2022 (Marinou et al., 2023). The region of interest (RoI) includes the broader North Atlantic Ocean, the Caribbean Sea, and the Western Saharan Desert, spanning latitudes from 0° to 45°.

The present article is structured as follows. In Sect. 2, the observational and model-based datasets (spaceborne, reanalysis, and ground-based) utilized towards the development and validation of the L2A+ dust product are outlined. In Sect.3, the methodology followed towards the detection and elimination of the cloud-contaminated profiles, the identification of the pure-dust layers using POLIPHON technique, and the derivation of the final pure-dust L2A+ backscatter, extinction, and mass concentration at 355 nm profiles are presented. The new Aeolus dust product (L2A+) is presented and discussed in Sect. 4, focusing on an indicative satellite overpass in proximity to Cabo Verde/Mindelo station on the 3rd of September 2021. Accordingly, quality assessment of the L2A+ dust product against ground-based reference measurements from eVe and Polly[XT] lidars is provided and discussed in Sect. 5, both in terms of specific cases of high interest and in addition, on the basis of all Aeolus-ALADIN and ground-based lidar validation concurrent measurements realized in the framework of the ESA-ASKOS experimental campaign. Finally, Sect. 6 presents and summarizes the main findings and conclusions.

## 2. Datasets

For the development of the refined Aeolus dust product, a series of processing steps has been undertaken, including the parallel use of polar-orbiting satellite and geostationary observations, in synergy with reanalysis numerical outputs and ground-based observations. In the current section, we will provide a thorough overview of all the data sources employed in this analysis, detailing their spatial and temporal resolutions, measurement principles, and specific roles in the development of the improved Aeolus dust product.

### 2.1 Aeolus/ALADIN aerosol optical products

The European Space Agency's (ESA) satellite wind mission, Aeolus, was launched on August 22, 2018, and operated until July 28, 2023. Its primary scientific goals were to enhance weather forecasting capabilities and deepen our understanding of atmospheric dynamics, including its interactions with the atmospheric energy and water cycles. A more detailed description of the Aeolus wind mission can be found in the Atmospheric Dynamics Mission-Aeolus (ADM-Aeolus) science report (ESA, 2008). Aeolus carried the ALADIN, the first space-based HSRL lidar which provided wind and particulate vertically resolved retrievals along the line-of-sight (LOS) directed at 35° off nadir (Flament et al., 2021). The instrument emitted 20 consecutive pulses of a circular polarized light at 354.8 nm, with a 50.5 Hz repetition frequency and received the co-polarized backscatter from molecules and particles or hydrometeors in two separate channels, referred to as the Mie and Rayleigh channels (Flamant et al., 2008b; Flament et al., 2021). A main difference between the two optical channels was that the Mie channel primarily detected the spectrally narrow return from atmospheric hydrometeors, while the Rayleigh channel detected the spectrally broader backscatter from atmospheric molecules (Dabas et al., 2008). A total number of 20-pulse accumulated signals were then transmitted to the ground yielding one measurement of ~3km horizontal resolution. During the on-ground data processing, a number of 30 measurements were further accumulated to form an "observation" or a "basic repeat cycle" (BRC) as it is called, corresponding to a distance of ~ 90 km along ALADIN's orbit-path. The detected signals were also vertically integrated in 24 height bins with a varying resolution that ranged from 250 m to 2 km depending on the range bin settings (RBSs) (Flament et al., 2021; Gkikas et al., 2023). Thin bins were preferable close to the ground, while at higher altitudes, thicker bins were needed due to the low density of molecules which in turn decreased the molecular backscatter coefficient.

The ALADIN's HSRL capability enabled the independent estimation of the volume extinction coefficient and co-polarized volume backscatter coefficient at 355 nm from the Mie and Rayleigh spectral channels, allowing a direct determination of the lidar ratio. However, this required robust crosstalk corrections to separate the molecular and particle signals (Gkikas et al., 2023). To exploit this capability, a specific algorithm, the Standard Correct Algorithm

(SCA), was designed for Aeolus, producing the Level-2A (L2A) product. The L2A product was derived from Mie and Rayleigh signals, factoring in instrument calibration constants, cross-talk coefficients which account for the imperfect separation of molecular and particulate spectra between the two HSRL channels, as well as laser pulse energy, accumulated pulses, and the molecular and particulate contributions to the measured signals (Flament et al., 2021).

These corrections yielded vertically resolved backscatter and extinction coefficients. A complete description of the main features included in the SCA algorithm can be found in Flament et al. (2021). While the derivation of the particulate backscatter coefficient was straightforward, for the extinction, the derivation was done via an iterative process from the top of the profile to the bottom applying a normalization function which used the measured and simulated pure molecular signals, under the assumption that the particle extinction at the topmost bin was zero.

However, this consideration made the SCA optical property products extremely sensitive to noise in the first bin (~20-25km), which was used as reference for the normalization, particularly under low signal-to-noise ratio (SNR) conditions due to the low molecular density at high altitudes. The SCA also produced the SCA middle-bin (SCA-MB) backscatter and extinction coefficient profile products by averaging the SCA neighboring vertical bins at a coarser resolution so as to reduce the noise in scenes with low SNR, thus obtaining a more stable product (Baars et al., 2021;

Dai et al., 2022).

As the L2A processor version evolves new algorithms have been also developed, aiming to address identified challenges related to the SCA algorithm. As such, a physical regularization scheme, namely the Maximum Likelihood Estimation (MLE), has been implemented within the L2A processor v3.14, to reduce the noise contamination of the SCA optical product (Ehlers et al., 2022). Improvements of the MLE algorithm include, among others, the introduction

of positivity and lidar ratio constraints that result into availability of particle extinction retrievals provision under the condition of the particle backscatter availability within the atmospheric profile, and vice versa (Ehlers et al., 2022). The evaluation of the MLE optical products against collocated ground-based measurements has shown a noteworthy improvement with respect to the SCA and SCA-MB optical products, indicating that the MLE algorithm provides a more solid basis for the estimation of extinction coefficient, co-polar backscatter coefficient, and lidar ratio at 355 nm.

In addition, available algorithms implemented in the Aeolus L2A processor, outperforming the SCA approach, include the Aeolus feature mask (AEL-FM) and the aerosol profile retrieval algorithm (AEL-PRO). Both algorithms have been developed in the framework of the Earth Cloud, Aerosol and Radiation Explorer (EarthCARE) (Illingworth et al., 2015; Wehr et al., 2023) activities related to developments of the HSRL Atmospheric Lidar (ATLID) and have been adapted to Aeolus (van Zadelhoff et al., 2023; Donovan et al., 2024). AEL-FM, which is outlined in the next

section, provides a probability mask for the presence of atmospheric features, and more specifically of clouds, aerosols, clear-sky, in the ALADIN profiles across Aeolus orbit-path at the highest available horizontal resolution. AEL-PRO, similarly to the MLE approach, is an optimal estimation based forward modeling retrieval which delivers profiles of extinction and backscatter. AEL-PRO uses the feature mask retrievals to facilitate improvements in the signal averaging process and to avoid averaging over weak (e.g. thin-cloud and aerosol) and strong (e.g. cloud)

scattering regions. In addition to these developments, the Aeolus L2A processor also includes the MLEsub product, which applies the Maximum Likelihood Estimation retrieval at a higher horizontal resolution (∼18 km) by reducing the extent of horizontal averaging. The MLEsub algorithm was designed to preserve finer-scale aerosol and cloud variability while retaining the noise-reduction benefits of the MLE approach. A recent validation study by Trapon et al. (2025), using airborne and ground-based lidar measurements collected during the Joint Aeolus Tropical Atlantic

Campaign (JATAC) above Cabo Verde in September 2022, demonstrated that the MLEsub optical profiles outperform the noisier SCA products. In particular, MLEsub showed improved robustness in conditions affected by signal attenuation, cloud contamination, and horizontal aerosol inhomogeneity, further reinforcing the advantages of the MLE-based approaches within the L2A product suite.

The primary Aeolus dataset used in our analysis covers the entire period of ASKOS operations at Mindelo, in Cabo

Verde (July, September 2021 - June, September 2022) and it consists of the raw backscatter coefficient profiles at the wavelength of 355 nm. These raw backscatter profiles correspond to the Aeolus L2A product, produced by the four different retrieval algorithms: the Standard Correct Algorithm (SCA), the SCA-MB (middle-bin) algorithm which provides the Aeolus L2A optical products by smoothing two consecutive vertical bins (23 instead of 24 vertical bins), the Maximum Likelihood Estimation algorithm (MLE), and the Aerosol Profile retrieval algorithm (AEL-PRO). The

datasets used in the framework of the present work were generated by the L2A 3.16 processor version, which corresponds to baseline 16.

### 2.1.1 Aeolus Feature Mask retrieval algorithm (AEL-FM)

The Aeolus classification product, called AEL-FM, provides at the highest available horizontal resolution a feature detection probability index with values ranging between 0 (clear sky) to 10 (likely very thick clouds) through the exploitation of the two-dimensional time-height correlation of the observational datasets. The mask does not distinguish between different particle types but instead, it detects areas of strong and weak returns or those associated with clear-sky conditions. AEL-FM is based on a median-hybrid method for the detection of strong features (Russ,
2006) and on a data smoothing strategy on the basis of a simplified maximum entropy method for the weaker ones (C. Ray Smith, 1985). Through this approach, AEL-FM enables the retrievals to deal with the low signal-to-noise ratio at a single pixel level. Table 1 provides the main classification output of the AEL-FM product with the first column of the table showing the feature detection probability indices with values ranging from -3 to 10 and the second column the definitions attributed to each feature index. Based on the definitions, clear-sky conditions labeled with a feature
index value of "0" are associated with very low signals that are likely to have been originated by clear air while stronger signals with values ranging from 6 to 10 are most likely to have originated from liquid or optically thick ice clouds. Additionally, the algorithm identifies regions on which the lidar beam has been fully attenuated (-2) and also identifies the surface returns (-3), in cases when the measured lidar backscatter signals are impacted by the surface. Figure 1 illustrates the retrieved Aeolus Feature Mask product for an indicative Aeolus overpass in the proximity of
Mindelo-Cabo Verde on the 3rd of September 2021 (orbit id: 017568). The Feature Mask output is provided at the Aeolus measurement scale of about ~3 km horizontal resolution. According to AEL-FM, the presence of different atmospheric features along the specific Aeolus track are evident, classified either as clouds or optically thick aerosol layers or those associated with clear sky conditions. In this particular atmospheric scene, one can clearly distinguish a partly attenuated area in the latitude band from 30 to 35˚N covered by a thick ice cloud between 9 and 15 km altitude.
Broken and low-altitude clouds of limited spatial extension with 'strong' return signals are also present throughout the entire Aeolus orbit-track. The AEL-FM algorithm is currently included in the latest Aeolus processor version of baseline 16, used in our analysis for the discrimination and elimination of the cloud-contaminated measurements along each Aeolus orbit.

| AEL-FM Index | Definition |
| --- | --- |
| 10 | Clouds |
| 9 | Most likely clouds |
| 8 | Very likely clouds or aerosols |
| 7 | More likely clouds or aerosols |
| 6 | Likely clouds or aerosols |
| 5 | Expected low altitude aerosol |
| 4 | Unlikely clouds or aerosol |
| 3 | Likely only molecules |
| 2 | Very likely only molecules |
| 1 | Most likely only molecules |
| 0 | Clear sky |
| -1 | Fully Rayleigh attenuated |
| -2 | No retrievals |
| -3 | Surface data |


**Table 1:** Aeolus feature-mask product definition. The first column provides the feature detection probability index ranging from -3 to 10. The second column shows the definition for each index.

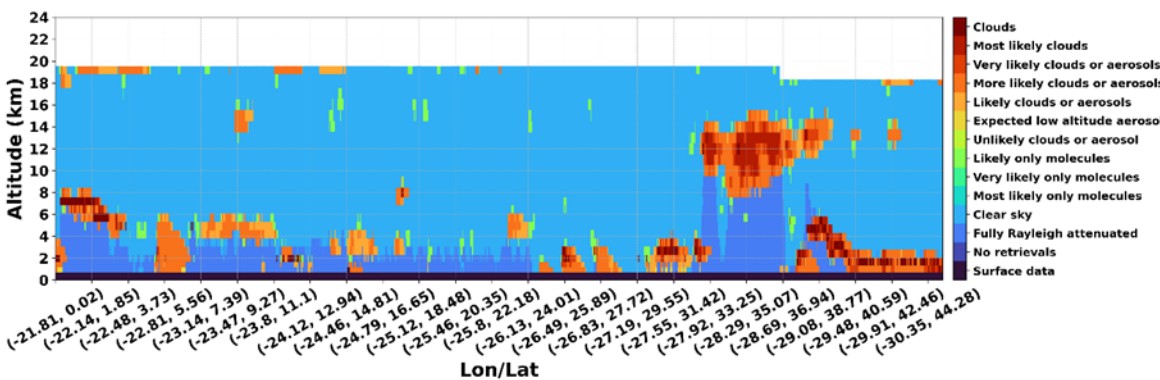

**Figure 1:** The high-resolution AEL-FM feature mask product for the Aeolus overpass on 3 September 2021 (orbit id: 017568).

## 2.2 SEVIRI CLAAS-3 data record


The Cloud property dataset (CLAAS), produced by the Satellite Application Facility on Climate Monitoring (CM SAF), a consortium created by the European Organization for the Exploitation of Meteorological Satellites (EUMETSAT), provides information on specific cloud properties, including among others the cloud-top pressure and temperature, cloud optical thickness, and cloud effective radius. The CLAAS-3 record, the latest version of CLAAS,
is generated based on measurements of the Spinning Enhanced Visible and Infrared Imager (SEVIRI) sensor onboard Meteosat Second Generation (MSG) satellites (Meirink et al., 2013). A detailed description of the CLAAS-3 data record with its previous editions can be found at Benas et al. (2023) and Stengel et al., (2014). CLAAS-3 provides, among other cloud properties, a binary cloud mask–cloud fractional coverage group which includes parameters concerning the initial cloud detection such as probabilistic cloud mask, binary cloud mask and cloud fractional
coverage. The data are available on multiple processing levels, starting from the level 2 variables which are provided every 15 minutes at the native SEVIRI spatial resolution of 3 km (nadir) and the level 3 retrievals, which provide spatiotemporal averages of the level 2 data such as daily averages and monthly averages in a 0.05° regular grid, as well as monthly diurnal cycle averages at a 0.25° grid resolution.

Aiming to achieve an optimum cloud-screening of the Aeolus optical product profiles over the study domain, the
SEVIRI CLAAS-3 Cloud Mask binary dataset, in synergy with AEL-FM product (sect.2.1.1.) is utilized. The horizontal resolution is about 4x4 km², depending on location in SEVIRI's field of view, and it provides the cloud mask product in 15 minutes temporal resolution and for the geographical region confined between 60°S and 60°N and between 60°W and 60°E. Figure 2 provides an indicative example of the cloud mask product output for the complete SEVIRI disc and for one time step on 17th September 2021 at 09:30 UTC, indicating the clear-sky and cloud-
contaminated areas in blue and grey, respectively.

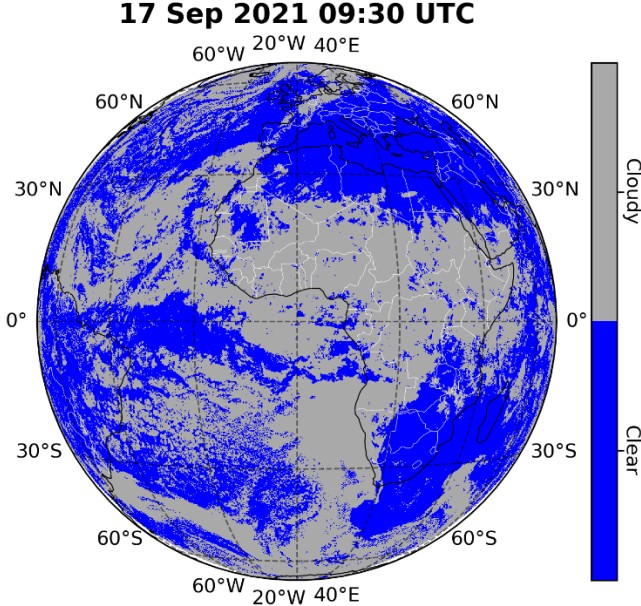

**Figure 2:** CLAAS-3 pixel-based cloud mask product for a given time step on 17th September 2021 (09:30 UTC).

**2.3 CAMS reanalysis dataset**

The CAMS reanalysis dataset, produced by the Copernicus Atmosphere Monitoring Service, is the latest global reanalysis data set of atmospheric composition (AC) and it consists of 3-dimensional time-consistent AC fields, including aerosols, chemical species and greenhouse gases. The reanalysis methodology incorporates satellite

observations with model outputs into a globally complete and consistent dataset using the European Centre for Medium-Range Weather Forecasts' Integrated Forecasting System (IFS) (Agustí-Panareda et al., 2023). The data set covers the period from January 2003 to June 2022 and it builds on the experience gained during the production of the earlier MACC reanalysis and CAMS interim reanalysis (Inness et al., 2013; Flemming et al., 2017). For the production of the CAMS reanalysis dataset, satellite retrievals of total column carbon monoxide (CO), tropospheric

column nitrogen dioxide (NO2), AOD; and profiles of ozone (O3) retrievals were assimilated (Inness et al., 2019). Compared with the previously produced CAMS interim reanalysis, the new ECMWF Atmospheric Composition Reanalysis (EAC4) has an increased horizontal resolution of ~80 km and in addition, it provides an increased number of chemical species at a higher temporal resolution (3-hourly analysis fields, 3-hourly forecast fields and hourly surface forecast fields) (Inness et al., 2019).

In this study, the absence of depolarization measurements in the Aeolus ALADIN lidar is addressed by integrating CAMS reanalysis data into the L2A+ workflow to classify aerosol types, separate dust from non-dust fractions, and provide the missing depolarization ratios required for the POLIPHON technique, enabling a more reliable identification of pure-dust layers. More specifically, reanalysis gridded CAMS outputs are utilized, with focus on the entire Region of Interest (RoI) and during the 4-month period of the ASKOS experimental campaign. The downloaded

dataset is characterized by a horizontal resolution of 1°, 60 hybrid sigma-pressure model levels on the vertical scale, and a temporal resolution of 3h. Aerosol species from CAMS are originally available in mass mixing ratio (kg/kg) and they include twelve prognostic tracers, consisting of three bins for sea salt grains of different sizes (0.03–0.5, 0.5–5 and 5–20 μm); three bins for dust (0.03–0.55, 0.55–0.9 and 0.9–20 μm); hydrophilic and hydrophobic organic matter and black carbon; and sulfate aerosols plus its precursor trace gas of sulfur dioxide (SO2) (Morcrette et al., 2009; Ryu

& Min, 2021). Conversion between the total aerosol mass mixing ratio (kg/kg) and mass concentration (μg/m3) is performed according to Eq.1:

$$C_{PM10} = \left( \frac{X_{SS1} + X_{SS2} + X_{SS3}}{4.3} + X_{DD1} + X_{DD2} + X_{DD3} + X_{OM} + X_{BC} + X_{SU} \right) * \left( \frac{p_m}{R_{spec} * T} \right) \quad (1)$$

where Xp denotes the mixing ratio of the aerosol type p, with $SS_{1,2,3}$ representing sea salt particles of different size classes, $DD_{1,2,3}$ representing dust particles, OM denoting organic matter, BC denoting black carbon and SU representing sulfate aerosols. Additionally, pm refers to the air pressure at the vertical layer midpoint (Pa), T the temperature at vertical layer midpoint (K), and Rspec = 287.058 J/(kg*K) is the specific gas constant for dry air. In the above formula (Eq. 1), the factor inside the parenthesis $\left( \frac{p_m}{R_{spec}*T} \right)$ represents the air density (kg/m³) derived from the ideal gas law. Multiplying the aerosol mass mixing ratio by the air density yields the mass concentration of the respective aerosol species in kg/m³, which is then converted to µg/m³ by scaling appropriately. This approach allows the calculation of the mass concentration of sea salt ($SS_1$, $SS_2$, $SS_3$), dust ($DD_1$, $DD_2$, $DD_3$), and smoke (OM + BC + SU) aerosols as required for the current study. As detailed in Section 3.3, the backscatter coefficients for total (dust, marine, and smoke ) and only-dust aerosols—used to estimate the particulate depolarization ratio values based on the POLIPHON equation—are then obtained from CAMS by first converting aerosol mass concentrations to extinction coefficients using POLIPHON extinction-to-volume conversion factors at 355 nm, and subsequently applying the appropriate lidar ratio values according to the recent literature.

## 2.4 Ground-based retrievals from eVe and PollyXT lidars

To calibrate and validate the primary (L2A) and refined (L2A+) Aeolus dust products under intense dust loads in the tropics, quality-assured reference ground-based measurements from eVe (Paschou et al., 2022) and Polly[XT] (Engelmann et al., 2016) lidars were collected during the Joint Aeolus Tropical Atlantic Campaign (JATAC) - ASKOS (Marinou et al., 2023), conducted in the Cabo Verde Islands in 2021 and 2022.

The eVe lidar, operated by NOA, is the ESA's ground reference system for the Aeolus products validation, specifically designed to provide the Aeolus mission with ground reference measurements of the optical properties of aerosols and thin clouds. A detailed description of the eVe lidar system is given by Paschou et al. (2022). In brief, eVe is a combined linear/circular polarization lidar system with Raman capabilities that operates at 355 nm and provides profiles of the particle backscatter and extinction coefficients, the lidar ratio, and the linear and circular depolarization ratios (Paschou et al., 2022, 2023). It is designed to be a mobile and flexible lidar system and it is implemented in a dual-laser/dual-telescope configuration that can point at multiple azimuth and off-zenith angles allowing eVe to reproduce the operation and pointing geometry of any linear or circular polarization lidar (space- or ground-based). As such, eVe can simultaneously reproduce the operation of the ALADIN lidar onboard Aeolus which uses circularly polarized emission at 355 nm as well as the operation of a traditional linear polarization lidar system (e.g. EARLINET). Additionally, to the referenced products, eVe is able to directly retrieve the Aeolus-like backscatter coefficient and Aeolus-like lidar ratio which are the reference ground-based lidar products that can be used for the assessment of the primary Aeolus products (Paschou et al., 2022).

The Polly[XT] lidar was provided by TROPOS and it is an automated multiwavelength Raman polarization lidar (Engelmann et al., 2016), designed to measure the aerosol loads in the boundary layer and the free troposphere. This specific lidar system enables measurements of the elastic backscatter coefficient at 355, 532 and 1064 nm, the inelastic backscatter at 387, 607 and 1058 nm, the cross-polar signal at 355, 532 and 1064 nm, and the inelastic signal from water vapor at 407 nm. For the 355 and 532 nm elastic channels as well as the 387 and 607 nm Raman channels in addition to far-field measurements, near-field measurements are available as well. The microphysical properties of liquid water droplets can also be determined, due to the dual-field-of-view depolarization channel (Jimenez et al., 2020a; Jimenez et al., 2020b). A detailed description of the lidar system, including error characterization can be found in Gebauer et al. (2024).

## 3. Methodology

The present section provides a step-by-step description of the methodology used to develop the new Aeolus dust product (L2A+). Figure 3 presents this methodology in a flowchart, outlining the procedures involved in the Aeolus L2A+ product development. According to the flowchart, the steps followed for the derivation of the refined Aeolus dust product include, among others, the use of the raw (unprocessed) Aeolus L2A optical products provided at the Aeolus observational scale (BRC level), the cloud-filtering approaches based on the synergistic use of the AEL-FM

dataset from the L2A processors II (Sect. 2.1.1) and the Cloud Mask product from SEVIRI (Sect. 2.2) for the derivation of the cloud-free aerosol profiles along each satellite overpass, the assignment of aerosol typing and the implementation of the one-step POLIPHON technique (Tesche et al., 2009; Mamouri & Ansmann, 2014; Mamouri & Ansmann, 2017; Ansmann et al., 2019) for the discrimination of the pure-dust layers based on reanalysis numerical outputs from CAMS (Sect. 2.3), as well as the use of several conversion formulas, including appropriate conversion

factors, for the correction of the backscatter coefficient and the final derivation of the L2A+ extinction and mass concentration profiles. Finally, the performance of the new Aeolus dust product has been validated against ground-based measurements acquired during the ASKOS/JATAC experiment at the Cabo Verde/Mindelo campaign site.

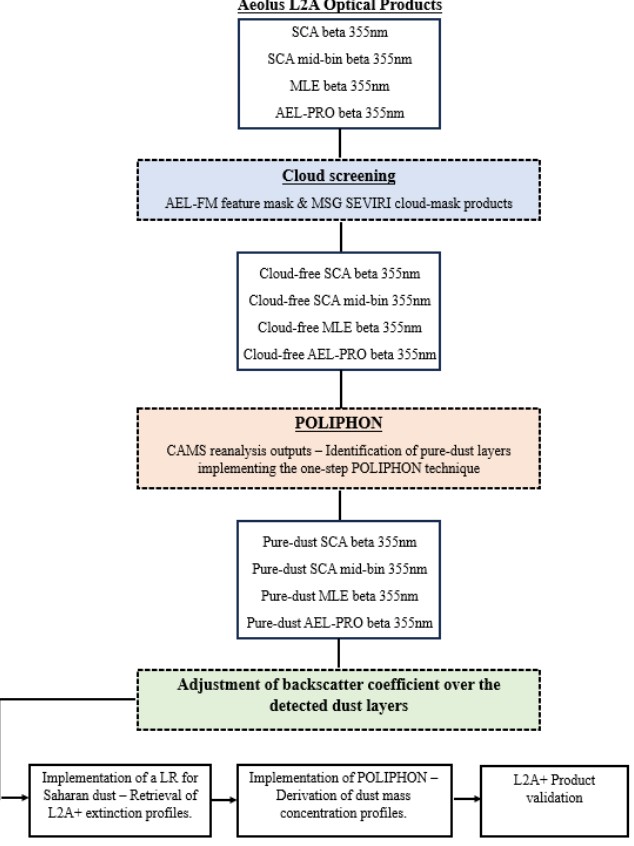

**Figure 3:** Flowchart of the L2A+ dust product development procedure.

A detailed description of all the processing steps followed until the final L2A+ dust product production is provided in the current section, focusing on an indicative Aeolus overpass close to Mindelo station, in Cabo Verde on the 3rd of September, 2021. Figure 4 illustrates, for the referenced time period, the Aeolus orbit-track over the study domain (blue thick line), along with the time-nearest spatial distribution of Dust Optical Depth over the entire study domain derived fromCAMS numerical outputs. According to the figure, we can see that the referenced satellite scanning track

coincides with a Saharan dust outbreak, when strong winds carried a thick plume of dust from N. Africa across the Canary Islands and across the Atlantic.

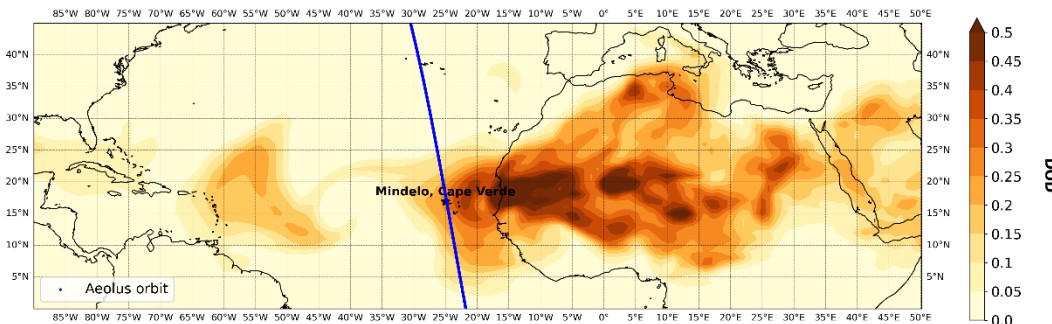

**Figure 4:** Spatial distribution of CAMS Dust Optical Depthover the entire study domain for an indicative case study on 3rd September 2021 (21:00 UTC). The blue line represents the time-nearest Aeolus ascending orbit over Mindelo, Cape Verde (orbit id: 017568).

### 3.1 Retrieval of the raw Aeolus L2A optical products

Figure 5 displays for the illustrated satellite overpass of Figure 4, the retrieved SCA (a), SCA-MB (b), MLE (c), and AEL-PRO (d) co-polarized (L2A) backscatter coefficient profiles at a horizontal resolution of ~90 km (one BRC) along the Aeolus orbit track. These are the raw Aeolus L2A backscatter coefficients, as retrieved from the four processor algorithms, without the application of any quality-assurance filtering. Although Aeolus quality-assurance flags are available for the L2A SCA and MLE products, they are not applied in the present study, as incorporating this additional flagging in the statistical analysis would lead to a substantial reduction in the number of valid bins per profile per BRC available for comparison. Regarding the AEL-PRO algorithm, the retrieved backscatter coefficients are initially provided at a fine horizontal resolution of approximately 3 km. Figure 5d presents the AEL-PRO backscatter coefficient profiles at the standard horizontal resolution of Aeolus L2A products (BRC scale), obtained by averaging 30 consecutive measurements that make up one BRC. According to the results, background noise patterns can be noticed in SCA and SCA-MB backscatter coefficients, while in the case of the MLE algorithm, we can observe more homogenous backscatter coefficients along track mostly attributed to the implemented constraints in the optical property retrievals for the specific algorithm. However, despite the noise in the retrieved SCA and SCA-MB co-polarized backscatter coefficients, it seems that both algorithms manage to capture a thick aerosol plume in the latitudinal band 2° to 22° N and between 2 and 6 km altitude. This aligns well with the spatial extent of increased Dust Optical Depth, as indicated by CAMS results (Figure 4), despite the 1 hour and 30 minutes time difference between the satellite and CAMS observations. Similarly, the plume can also be detected by the MLE and AEL-PRO retrieval algorithms in the specific latitude/altitude range with considerably lower noise patterns in the backscatter estimates. We can also notice some BRC bins with high backscatter coefficient values at 355 nm (bright yellow, orange). These high values are primarily caused by the presence of clouds in these specific regions. Moreover, in the case of optically thick clouds, the signal has been fully attenuated, resulting in no measurements' acquisition below the detected cloud layers. BRC bins with a strong presence of clouds are detected and eliminated from the analysis so as to retrieve the cloud-free aerosol profiles at each satellite overpass.

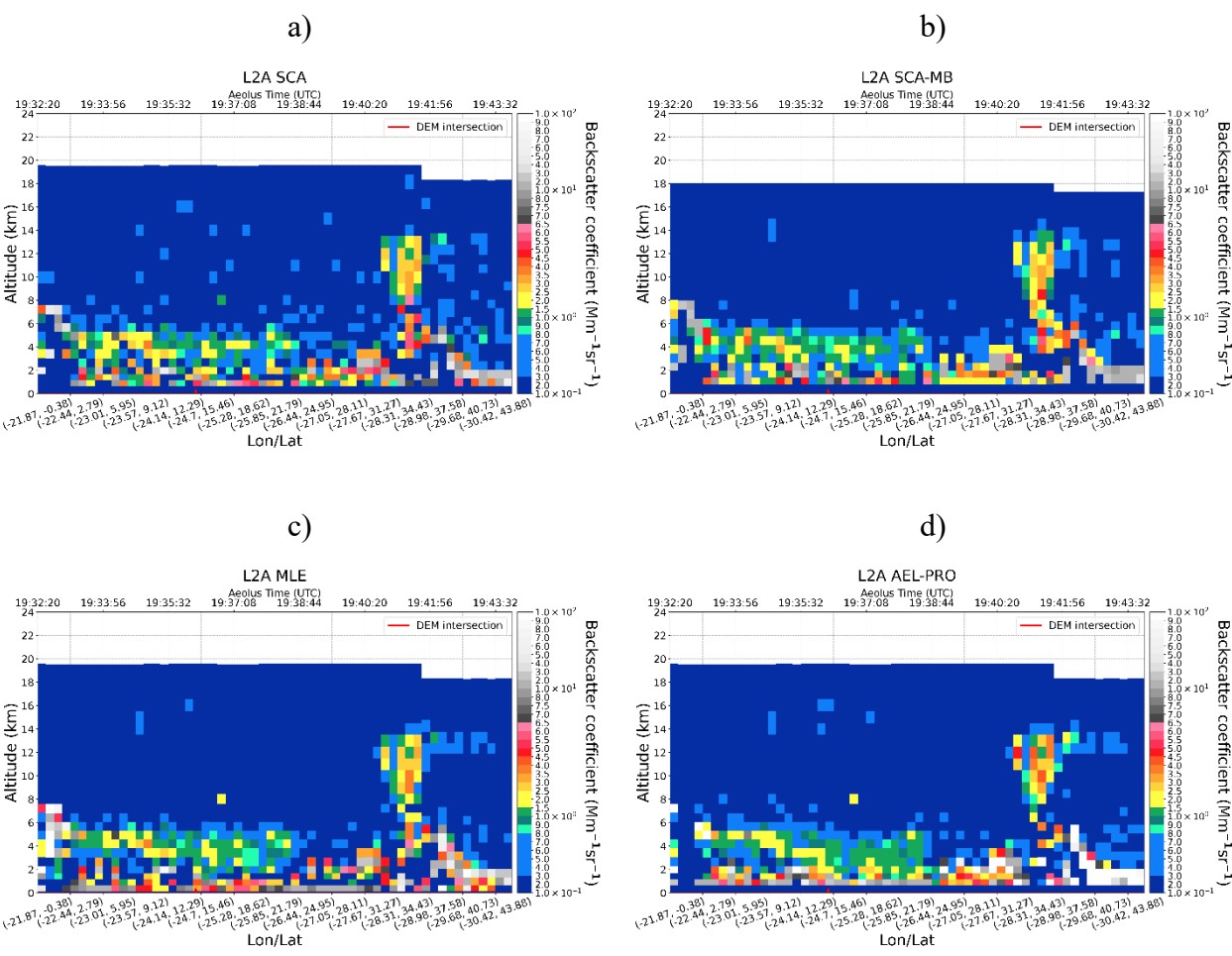

**Figure 5:** Retrieved co-polarized (L2A) backscatter coefficient cross-sections measured at 355nm for the SCA (a), SCA-MB (b), MLE (c), and AEL-PRO (d) processor algorithms along an indicative Aeolus overpass on 3 September 2021 (orbit id: 017568).

## 3.2 Cloud-screening method

This step of the methodology includes a rigorous filtering of the raw Aeolus L2A retrievals for the exclusion of all the cloud contaminated profiles from the final dataset that will be processed. Although the L2A product (Baseline 16) provides a built-in cloud flag for both the SCA and MLE retrievals (Flamant et al., 2022)—generated at the BRC level using ECMWF auxiliary meteorological fields, specifically the ice and liquid water content information—we opted not to rely solely on this product. Instead, we applied a combined cloud-screening approach using the Aeolus Feature Mask (AEL-FM) algorithm and the SEVIRI CLAAS-3 binary cloud-mask dataset to ensure a more robust and conservative identification of cloud-free conditions along each Aeolus track.

Based on AEL-FM, the features of the probed atmospheric scene are classified, at the finest available resolution, to those associated with "strong" and "weak" return signals mainly attributed to clouds or aerosols, respectively, and those from the molecular (Rayleigh) atmosphere. However, the implementation of the cloud filtering procedure based on the AEL-FM dataset is not straightforward. This complexity is largely due to the different horizontal resolutions of the AEL-FM and the Aeolus L2A retrievals. To be more specific, the Aeolus L2A retrievals are available at coarse horizontal scales that cover a horizontal distance of ~90km, whereas the primary AEL-FM dataset is provided at the Aeolus measurement resolution of ~3km. Therefore, prior to the implementation of the cloud-filtering procedure, a

common BRC bin is established between Aeolus L2A and AEL-FM data. This is achieved by aggregating those measurements residing within the margins of each Aeolus BRC bin, with associated feature index values ranging from
6 to 10. The specific indices were selected based on the definitions provided in Table 1, since they are most likely to have originated by cloud returns. It is worth mentioning here that depending on the L2A processor version, the total number of accumulated measurements in one BRC profile may vary. In this case, for the processor version 3.16, each BRC bin has 30 measurements. Figure 6 illustrates for the retrieved AEL-FM product of the orbit 017568, displayed in Figure 1, the reconstructed AEL-FM cloud dataset downgraded to the Aeolus observational horizontal resolution
(BRC level), which provides, separately for the regular (24 bins) and middle-bin (23 bin) vertical resolution of Aeolus, the total percent of cloud contaminated measurements within each BRC bin.

a)                                                                    b)

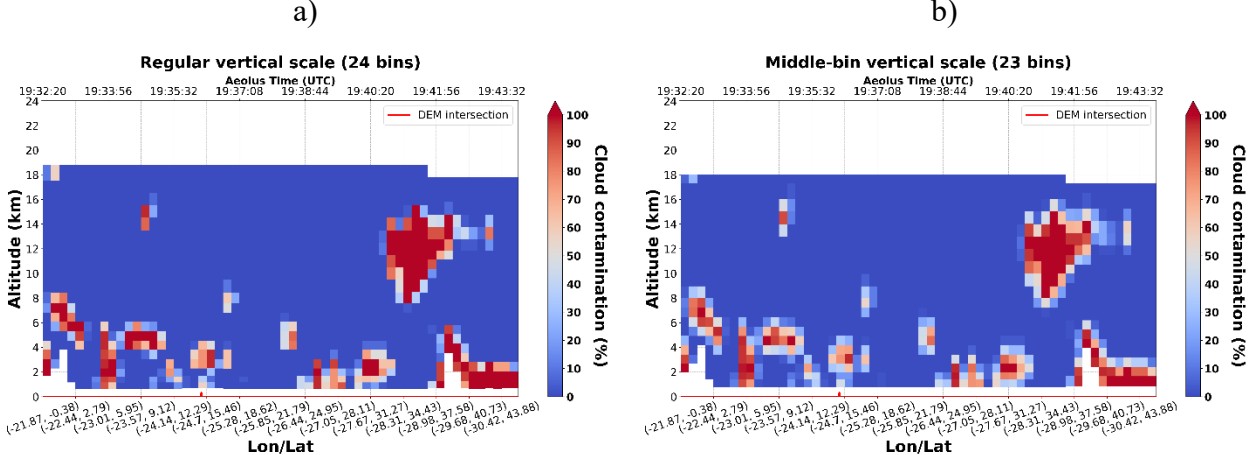

**Figure 6:** The transformed AEL-FM dataset to the Aeolus observation scale of ~90km horizontal distance for the a) regular (24 bins) and b) middle-bin (23 bins) vertical resolutions for the Aeolus overpass on 03 Sep 2021 (orbit id: 017568).

Based on the reconstructed AEL-FM product of Figure 6, in the final step of the filtering analysis, the BRC bins with frequency of occurrence of clouds exceeding the threshold value of 0% were excluded from the analysis with the associated observations from all the Aeolus L2A optical product retrievals. The aforementioned filtering analysis has been implemented separately to all the Aeolus L2A products provided at the regular vertical scale (SCA, MLE, and AEL-PRO retrievals) and those provided at the middle-bin scale (SCA-MB products).

The retrieved cloud-free Aeolus L2A products were also filtered out based on the SEVIRI CLAAS-3 dataset. Due to the high temporal and spatial resolution of the Cloud Mask product from SEVIRI, a very good temporal and spatial collocation with Aeolus can be achieved. In our case, the finest available Aeolus horizontal resolution of ~3 km has been selected for the collocation process with the Cloud Mask dataset which was achieved via a nearest-neighbor approach in both space and time, leading to maximum spatial and temporal distances not exceeding 3 km and 7.5 min,
respectively. Once collocation was carried out, the cloud fraction of each Aeolus BRC profile was binarized using a cloudiness threshold of 50%. This process facilitated the exclusion of specific BRC profiles with cloud fraction exceeding the applied threshold value. Figure 7 provides an example of the CLAAS-3 Cloud Mask product at the nearest timestep to the Aeolus overpass on 3 September 2021 where the grey-shaded areas represent the spatial coverage of clouds along the satellite's track.

The retrieval of the cloud free aerosol profiles is achieved through synergistic implementation of AEL-FM and CLAAS-3 along each Aeolus granule.

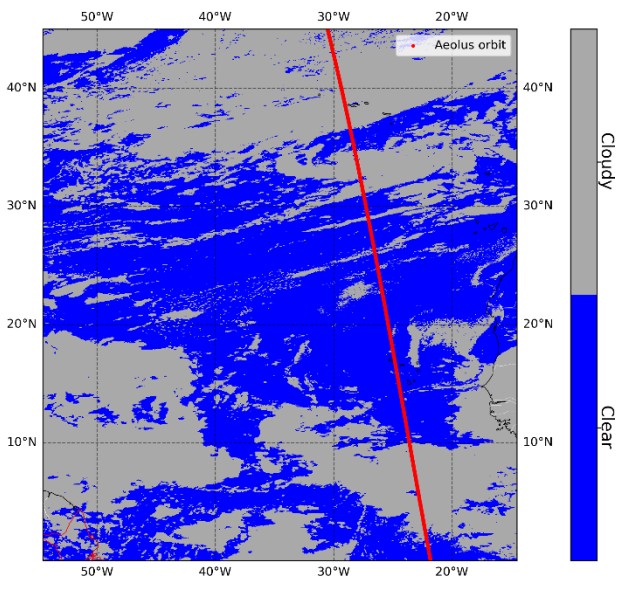

**Figure 7:** SEVIRI Cloud Mask product for a given time step on 3 September 2021 (19:30 UTC). The time-nearest Aeolus overpass on 3 Sep 2021 is also depicted with the start and end times (in UTC) of the ALADIN observations.

In Figure 8, the cloud-free Aeolus SCA (a), SCA-MB (b), MLE (c), and AEL-PRO (d) co-polarized backscatter profiles after combining both filtering procedures are displayed, for the case of the Aeolus overpass on 3 September 2021 in the Cabo-Verde Mindelo area. Individual BRC bins of high presence of clouds along the Aeolus track and an extensive area within the latitude range of 20° to 40° N which have been filtered out according to AEL-FM and SEVIRI cloud products, are evident. It is important to note that the lower vertical resolution of the SCA-MB product
increases the apparent thickness of the cloud layer, which in turn enlarges the filtered area. This leads to a significant reduction in the number of available cloud-free backscatter profile observations for the specific algorithm. In the case of the AEL-PRO algorithm, the retrieved cloud-free co-polar backscatter profiles were further filtered out based on the classification product provided by AEL-PRO, keeping only the BRC bins classified as aerosols (index =103 for tropospheric aerosols). Figure 8d illustrates the pure-aerosol co-polarized backscatter profiles retrieved from the AEL-
PRO algorithm after combining all the available cloud-filtering and classification tools.

a)                                                                                                    b)

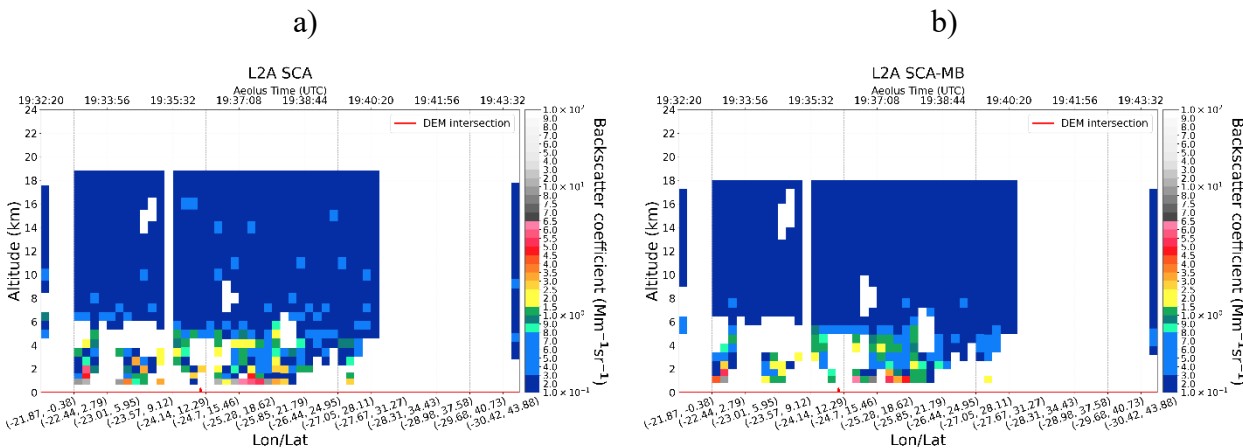

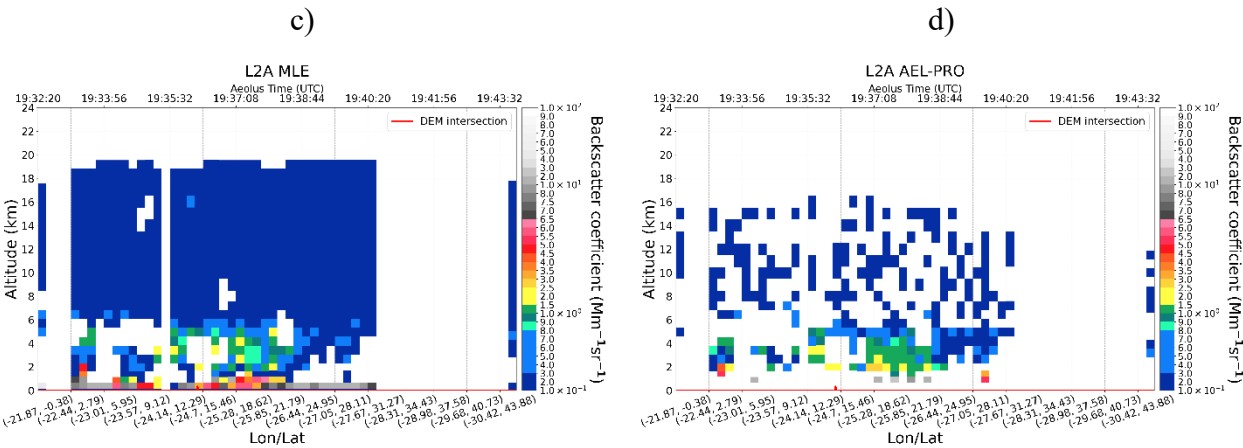

**Figure 8:** Cloud-free total co-polarized (L2A) backscatter coefficient cross-sections measured at 355nm for the SCA (a), SCA-MB (b), MLE (c), and AEL-PRO (d) retrieval algorithms along the Aeolus overpass on 3 September 2021 (orbit id: 017568).

### 3.3 Quantifying the Aeolus pure-dust component through reconstruction of its missing cross-polar term

The next step of the methodology involves the separation of the atmospheric dust contribution from the total aerosol load. This is achieved using the one-step POLIPHON technique, a well-established method developed within the framework of the European Aerosol Research Lidar Network (EARLINET) (Pappalardo et al., 2010). POLIPHON enables the decomposition of the aerosol mixture into dust and non-dust components by exploiting particle depolarization ratio measurements, which serve as a key indicator of the presence of non-spherical mineral dust particles (Proestakis et al., 2024). A limitation arises, however, due to the fact that the ALADIN lidar system does not provide particle depolarization ratio data. Such measurements are indispensable for discriminating between dust and non-dust aerosol components and for applying the POLIPHON approach in its original form. To overcome this constraint, we integrated CAMS reanalysis products into our workflow. Specifically, CAMS data were used to (i) provide information on aerosol type classification, thereby guiding the attribution of the detected aerosol load to either dust or non-dust fractions, and (ii) supply the missing depolarization ratio values required to implement the POLIPHON technique.

To identify Aeolus BRC bins associated with the presence of dust, the primary task consisted of performing a careful spatial and temporal collocation between the Aeolus L2A retrievals and the CAMS reanalysis products. In temporal terms, the time-nearest CAMS outputs with maximum time difference of ±3h from each Aeolus observation step were selected. Spatially, as in the case of the Aeolus-SEVIRI collocation procedure presented in Sect. 3.2, the nearest-neighbor technique was applied in order to extract the unique 1°x1° grid point of CAMS closest to each Aeolus observational step. Once the spatiotemporal collocation process was carried out, the averaged values of CAMS aerosol mass concentration retrievals, residing within the altitude margins of each Aeolus BRC both at the regular (24 bins) and middle-bin (23 bins) vertical scale, were computed. Figures 9a and 9b present the vertical cross-sections of the CAMS dust mass concentration and dust fraction (i.e., the ratio of dust to total aerosol mass concentration), respectively, provided at the same horizontal and vertical resolution as the Aeolus L2A optical products. Both parameters were employed in our analysis to identify atmospheric layers with a strong dust presence. Over these layers, the missing cross-polar backscatter component was adjusted in order to derive an improved total (L2A+) dust backscatter coefficient, along with the corresponding L2A+ dust extinction and mass concentration values. As an illustrative example, the Aeolus overpass on 3 September 2021 over the Cabo Verde–Mindelo region clearly highlights a prominent dust layer. The CAMS outputs reveal enhanced dust concentrations and consistently high dust fractions extending across the latitudinal band from 2° to 20° N and spanning altitudes between approximately 1 and 6 km. This structure is indicative of the well-developed Saharan Air Layer, frequently observed in this region, and provides an ideal test case for evaluating the capability of the L2A+ methodology to capture dust-dominated aerosol layers with improved accuracy.

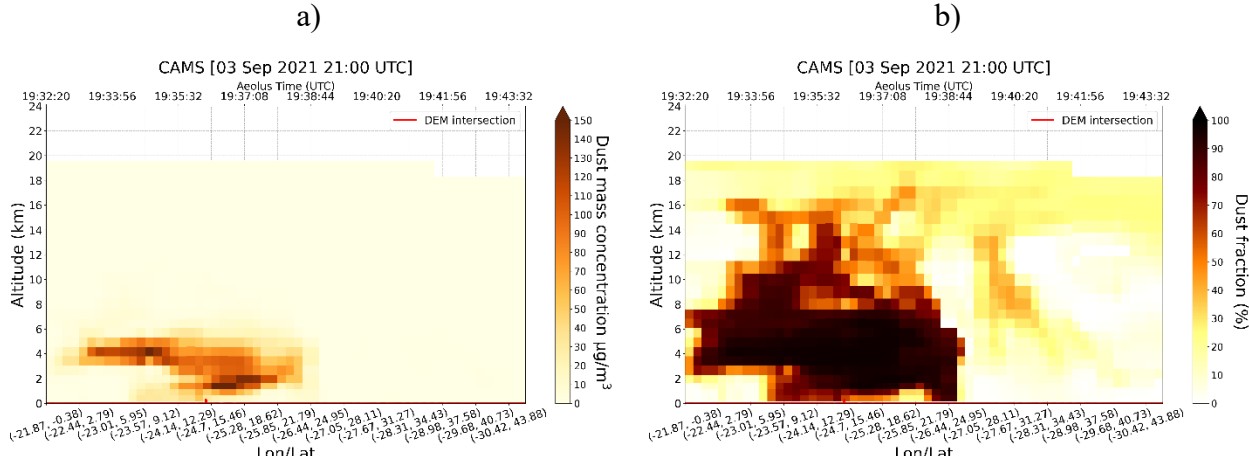

**Figure 9: a)** Dust mass concentration and **b)** dust fraction vertical cross sections along the Aeolus orbit over Mindelo, Cabo Verde (orbit id: 017568) at the regular vertical scale of Aeolus.


The next step in our analysis focuses on estimating the missing depolarization ratio values using CAMS products. In particular, by applying the POLIPHON equation (Eq. 2) for calculating the contribution of the pure-dust aerosol component to the total aerosol load in terms of the backscatter coefficient (Proestakis et al., 2024) and solving it with respect to the particle linear depolarization ratio, we derive the corresponding particle depolarization ratio value.


$$\beta_{\lambda,d}(z) = \beta_{\lambda,p}(z) \frac{(\delta_{\lambda,p}(z) - \delta_{\lambda,nd}(z)) (1 + \delta_{\lambda,nd}(z))}{(\delta_{\lambda,d}(z) - \delta_{\lambda,nd}(z)) (1 + \delta_{\lambda,p}(z))} \quad (2)$$

In Eq. (2), the parameters $\beta_{\lambda,d}$, and $\beta_{\lambda,p}$ denote the pure-dust and total backscatter coefficients respectively. The constants $\delta_{\lambda,d}$ and $\delta_{\lambda,nd}$ represent the characteristic particulate depolarization ratio of the pure-dust and non-dust components of the external aerosol mixture, expressed as functions of wavelength "λ" and height "z". Then, to estimate

the particle depolarization ratio from Eq. (2), it was first necessary to determine the pure-dust and total backscatter coefficients from CAMS. This was accomplished by calculating the mass concentration values of dust aerosols ($DD_{1,2,3}$) and non-dust species ($SS_{1,2,3}$, OM, BC, SU) using Eq. (1). The corresponding backscatter coefficients for each aerosol type (dust, marine, and smoke) were then retrieved through a two-step process: (i) conversion of aerosol mass concentration to extinction coefficient using the appropriate POLIPHON extinction-to-volume conversion factors at

355 nm for dust and marine aerosols (Mamouri and Ansmann, 2014) and for smoke aerosols (Baars et al., 2021); and (ii) application of lidar ratio values at 355nm from the DeLiAn database (Floutsi et al., 2023). It is important to note that, for smoke aerosols, the POLIPHON extinction-to-volume conversion factors are only available at 532 nm and not at 355 nm. Therefore, the retrieved backscatter coefficient at 532 nm ($\beta_{532}$) was converted to its 355 nm equivalent ($\beta_{355}$) using the color ratio CR = $\beta_{355}$/ $\beta_{532}$ as proposed by Veselovskii et al. (2025). Based on observational studies in

Lille (Hu et al., 2022), a typical CR value of about 2.2 was adopted for aged smoke. The pure-dust backscatter coefficient was first determined, and the total backscatter coefficient was then obtained by summing the contributions from the pure-dust and non-dust components (smoke and marine aerosols). Corresponding depolarization ratios $\delta_{355,d}$ and $\delta_{355,nd}$, were set to 0.244 for dust and 0.03 for non-dust aerosol species at 355 nm, based on the DeLiAn database(Floutsi et al., 2023). These backscatter and depolarization ratio values were then used as inputs in Eq. (2) to

calculate the CAMS-based particle linear depolarization ratio.

Finally, the resulting particle linear depolarization ratio $\delta_{355,p}$ was converted to its circular counterpart using Eq. (3).

$$\delta_{circ}^{355} = \frac{2\,\delta_{linear}^{355}}{1 - \delta_{linear}^{355}} \quad (3)$$

where $\delta_{linear}^{355}$ is the computed linear depolarization ratio at 355 nm from Eq (2). Figure 10 shows, for the selected scene, the particle circular depolarization ratio profiles at 355 nm derived from CAMS, scaled to the Aeolus horizontal and vertical BRC.

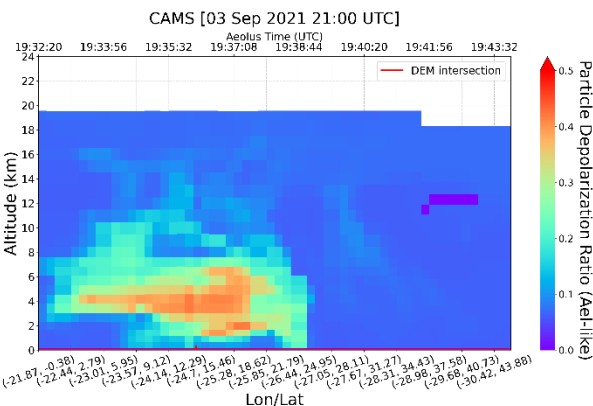

**Figure 10:** Particle circular depolarization ratio vertical cross sections along the Aeolus orbit over Mindelo, Cabo Verde (orbit id: 017568) at the regular vertical scale of Aeolus.

Based on the CAMS-retrieved particle depolarization ratio data, and considering the central parameters $\delta_{355,d}$ and $\delta_{355,nd}$, the one-step POLIPHON method enables the separation of pure-dust and non-dust components within an assumed external aerosol mixture. Specifically, mixtures with particle depolarization ratios lower than $\delta_{\lambda,d}$ and greater than $\delta_{\lambda,nd}$, can be decoupled into their respective contributions, while cases where $\delta_{\lambda,p}(z) \leq \delta_{\lambda,nd}(z)$ are considered as dust-free and cases where $\delta_{\lambda,d}(z) \leq \delta_{\lambda,p}(z)$ are regarded as consisting entirely of dust. By employing the formula for calculating the particle depolarization ratio, the missing Aeolus cross-polar backscatter signal at 355 nm can be reconstructed using the following relation:

$$\beta_{total}^{cross} = \delta_{circ}^{355} * \beta_{total}^{co} \quad (4)$$

where $\beta_{total}^{cross}$ represents the missing cross-polar backscatter contribution from both dust and non-dust aerosols ($\beta_{total}^{cross} = \beta_{dust}^{cross} + \beta_{non-dust}^{cross}$) and $\beta_{total}^{co}$ is the directly observed Aeolus co-polar backscatter from both aerosol types ($\beta_{total}^{co} = \beta_{dust}^{co} + \beta_{non-dust}^{co}$). In Eq. (4), the terms $\beta_{dust}^{cross}$, and $\beta_{non-dust}^{cross}$ can be expressed as functions of their respective co-polar counterparts $\beta_{dust}^{co}$, and $\beta_{non-dust}^{co}$ using the theoretical depolarization ratios of dust and non-dust components ($\delta_{355,d} = 0.244$, $\delta_{355,nd} = 0.03$), converted to circular values ($\delta_{355,d} = 0.65$, $\delta_{355,nd} = 0.06$).This formulation enables the estimation of the pure-dust and non-dust co-polar backscatter components, from which the corresponding missing cross-polar contributions can subsequently be derived. Finally, by adding the reconstructed pure-dust cross-polar term to the pure-dust co-polar contribution, the complete (L2A+) pure-dust backscatter coefficient can be obtained. Similarly, by accounting for the non-dust cross-polar component, the total Aeolus backscatter coefficient can be retrieved. This step is essential for aerosol characterization with Aeolus, as it restores the otherwise unavailable cross-polar information required to disentangle dust from non-dust backscatter signals.

**3.4 Retrieval of the L2A+ pure-dust extinction and mass concentration profiles**

In this phase, the new L2A+ pure-dust extinction coefficient at 355 nm profiles for dust aerosols were derived by multiplying the retrieved Aeolus total circular pure-dust backscatter coefficient at 355nm with an appropriate lidar ratio value of 53.5sr for Saharan dust obtained from the DeLiAn database (Floutsi et al., 2023). Finally, the retrieved L2A+ extinction coefficient at 355 nm profiles were converted to mass concentration profiles using the POLIPHON method (Tesche et al., 2009; Mamouri & Ansmann, 2014; Mamouri & Ansmann, 2017; Ansmann et al., 2019). The formula used for the retrieval of the mass concentration of dust is given by (Eq.4):

$$M_d = p_d * v_d \quad (5)$$

with the dust particle density $p_d$ of 2.6 g cm$^{-3}$ (Ansmann et al., 2012)and $v_d$ the dust volume concentration. The dust volume concentration in Eq. (5) can be estimated through the following conversion formula (Eq.6):

$$v_d = c_{v,d,532} * \sigma_{d,532} \quad (6)$$

which includes the particle extinction coefficient at 532nm ($\sigma_{d,532}$), and the extinction-to-volume conversion factor at 532nm ($c_{v,d,532}$) derived from the AERONET long-term observations (Ansmann et al., 2019). Since the above conversion formula uses the particle extinction coefficient retrieved at 532 nm, the Aeolus L2A+ pure-dust extinction profiles had to be converted at first from 355nm to 532nm following the well-known Ångström exponential law as follows:

$$\sigma_{\lambda2} = \sigma_{\lambda1} \left(\frac{\lambda_1}{\lambda_2}\right)^{A_{\frac{\lambda_1}{\lambda_2}}} \quad (7)$$

where $\sigma_{\lambda2}$ is the converted extinction coefficient at $\lambda_2$ = 532nm, $A_{\lambda1/\lambda2}$ is the extinction-related Ångström exponent and $\sigma_{\lambda1}$ is the dust L2A extinction coefficient of Aeolus at $\lambda_1$ = 355 nm. In the above formula (Eq.7), the extinction-related Ångström exponent of the order of 0.1 was selected for Saharan dust aerosol type according to the DeLiAn database presented in Floutsi et al. (2023).

## 4. Results

### 4.1 Presentation of the Aeolus L2A+ dust product during a dust transport episode on 3 September 2021

On 3 September 2021, as already described in Sect. 3, CAMS tracked a dust transport episode when Easterly trade winds transported a significant amount of dust particles from the Saharan Desert across the North Atlantic towards the Caribbean. Lying directly west of the Sahara, the Cabo Verde islands are frequently affected by these advected dust loads. At the specific period, as we can see in Figure 4, CAMS reported high levels of DOD over the islands with a significant impact on the air quality of the region. This is an ideal case study for developing and validating the refined Aeolus dust product at the specific time since depolarizing mineral particles are probed by ALADIN which is not able, in this case, to detect the cross-polar component of the backscattered lidar signal, underestimating the backscatter coefficient. Following the cloud-filtering and one-step POLIPHON pure-dust discrimination procedures, outlinedin section 3, the pure-dust backscatter profiles for all the Aeolus orbits over the entire study domain were derived. In Figure 11 we present for the specific case study, the Aeolus SCA, SCA-MB, MLE, and AEL-PRO pure-dust co-polarized (L2A) backscatter profiles at 355nm (left panel), along with the adjusted pure-dust total (L2A+) backscatter profiles at 355 nm. At first glance, the results clearly show an increase in the L2A+ backscatter coefficient for the detected dust layers following the adjustment of the mis-detected cross-polarized backscatter component, as described in Section 3.3. This enhancement highlights the effectiveness of the reconstruction methodology in recovering previously missing cross-polar information, allowing for a more accurate representation of the dust backscatter profile. Consequently, the refined profiles provide a more reliable basis for quantifying dust contributions and separating them from non-dust aerosol signals in the studied atmospheric scenes.

a)                                                                      b)

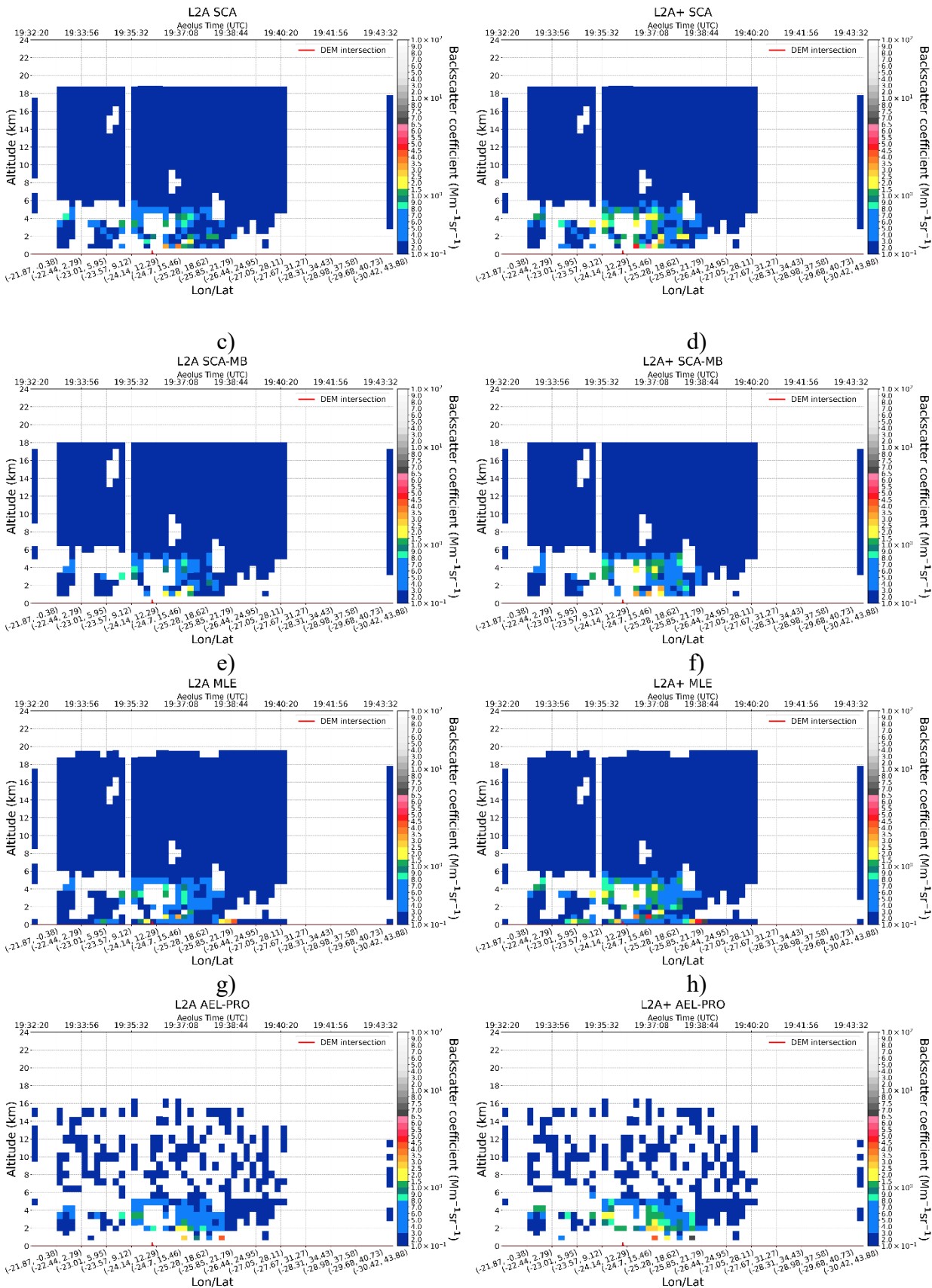

**Figure11:** Pure-dust L2A co-polar backscatter profiles (a, c, e, g) and pure-dust L2A+ (co + cross) backscatter profiles (b, d, f, h) retrieved from the SCA, SCA-MB, MLE, and AEL-PRO processor algorithms respectively for the Aeolus overpass on 3 September 2021 (orbit id: 017568).

Based on the retrieved pure-dust L2A+ backscatter profiles at 355nm, a lidar ratio for Saharan dust was applied to retrieve the new pure-dust L2A+ extinction profiles at 355nm and then, as already has been described in Sect. 3.4, the Ångström exponential law was used to convert the L2A+ extinction profiles from 355nm at 532nm before implementing the POLIPHON conversion formulas for the retrieval of the mass concentration profiles. The reconstructed pure-dust L2A+ extinction profiles at 355nm along with the final L2A+ dust mass concentration profiles are illustrated in Figure 12 separately for the four processor algorithms, SCA, SCA-MB, MLE, and AEL-PRO
respectively.

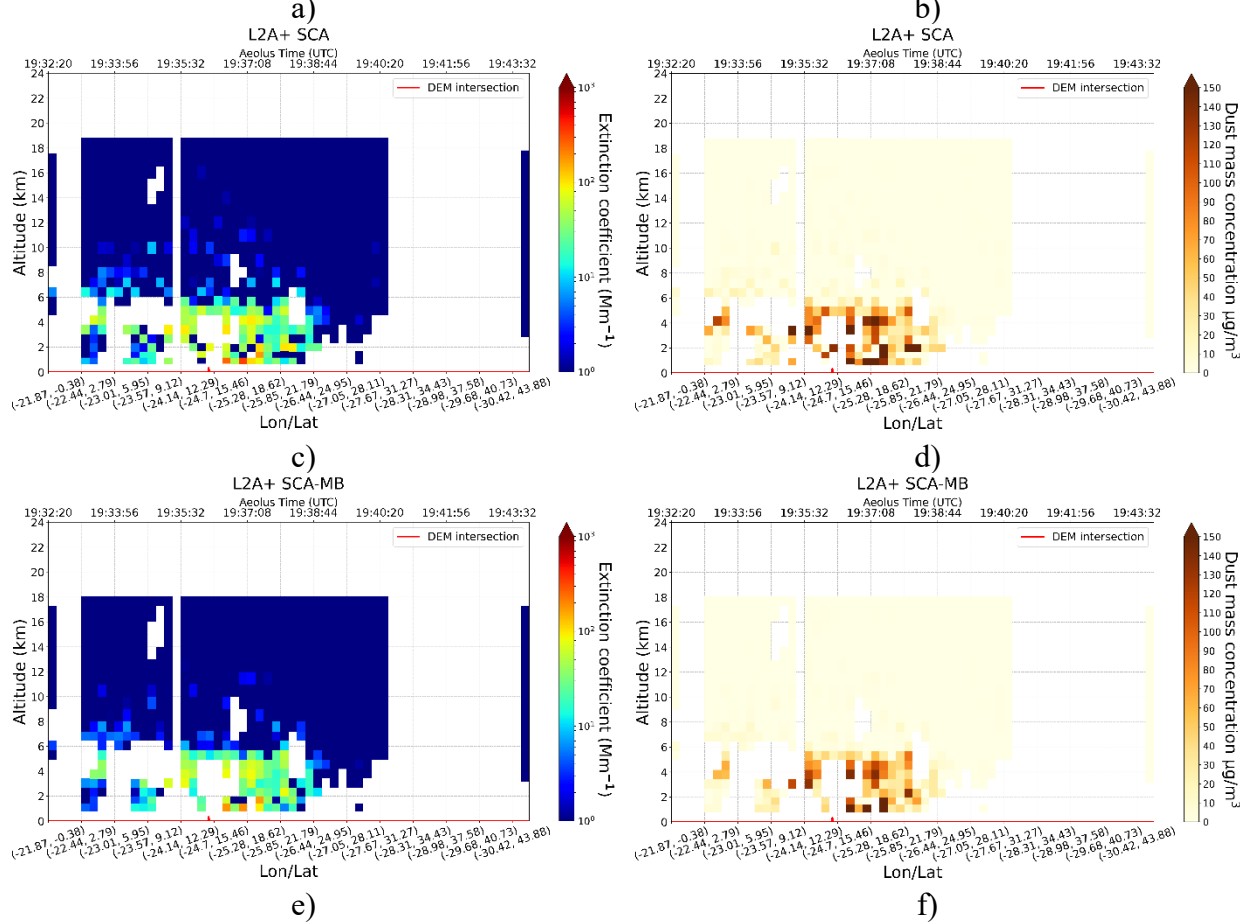

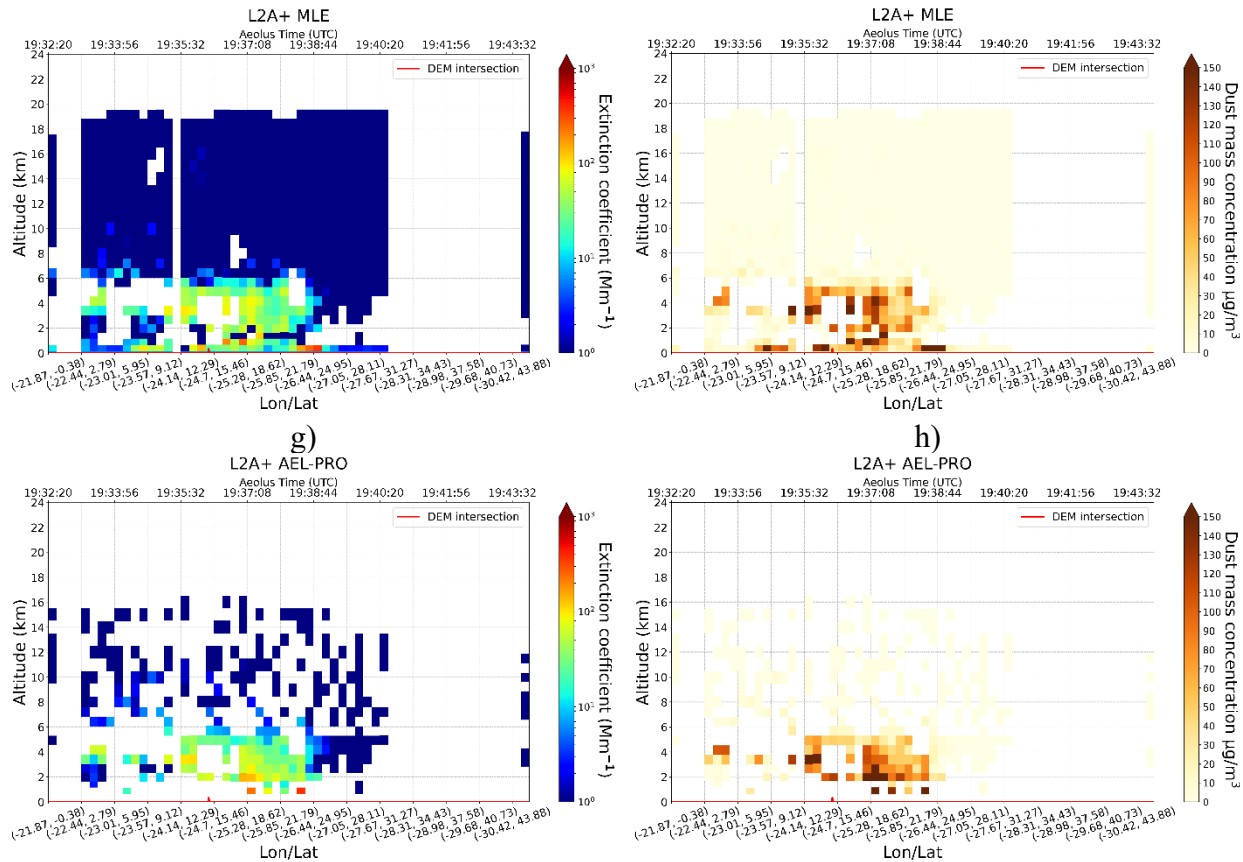

**Figure 12:** Pure-dust L2A+ extinction at 355nm (a, c, e, g) and mass concentration profiles (b, d, f, h) retrieved from SCA, SCA-MB, MLE, and AEL-PRO algorithms for the Aeolus overpass on 03 September 2021 (orbit id: 017568).

## 5. Validation of the Aeolus pure-dust L2A+ product

The validation results of the new pure-dust Aeolus L2A+ product against the corresponding ground-based retrievals, operated in the framework of the JATAC/ASKOS campaign in Cabo Verde, are presented in the current section. The validation procedure aims to provide confidence regarding the quality of the pure-dust L2A+ product and will drive possible adjustments on ALADIN observational capabilities necessary for minimizing the deviations found between ground-based and spaceborne profiles. The present section is divided into two parts. In the first part, we selected an indicative case study to compare individual profiles of the pure-dust backscatter coefficient from Aeolus with the co-located profiles of pure-dust backscatter coefficients acquired from the ground-based eVe and Polly[XT] lidars. In the
second part, a more robust assessment analysis was performed expanding the dataset to all the coincident Aeolus - ground-based observations through the entire period of the ASKOS/JATAC experiment at Cabo Verde. It is worth noting that the retrieval of pure-dust backscatter coefficient profiles from eVe was carried out using a method analogous to that applied for Aeolus, where Eq. (2) was employed to isolate and quantify the contribution of the pure-dust component.

### 5.1 Ground-based validation for the case study on 3 September 2021

Aeolus passed close to the ground-based monitoring station of Mindelo at 19:36 UTC on 3 September 2021. The selected case study presents the intercomparison process between satellite and ground-based observations and serves as a graphic example of the pure-dust Aeolus L2A+ product performance. Figure 13 shows the comparison of the Aeolus L2A and L2A+ step-like vertical profiles of the pure-dust backscatter coefficient at 355 nm as retrieved from the SCA, SCA-MB, MLE, and AEL-PRO algorithms against the corresponding backscatter coefficient at 355 nm profile acquired from the ground-based eVe lidar. For comparison purposes, the eVe backscatter coefficient at 355 nm profile is presented both at its original high vertical resolution (red) and at Aeolus's vertical resolution (magenta). The eVe backscatter coefficient profile is further provided at both the regular Aeolus vertical scale (24 bins) for SCA/MLE/PRO vs. eVe comparisons and the middle-bin scale (23 bins) for SCA-MB vs. eVe comparisons. As indicated in the figure's heading, the ground-based lidar backscatter observations were collected between 18:57 and 20:22 UTC. Furthermore, the lidar is located 36 km from the midpoint of the nearest Aeolus observation (which is ~90 km average along the satellite track). It should be noted that only the quality-assured, pure-dust ground-based measurements have been included in the comparison process. What we can first notice is that the Aeolus backscatter coefficients from the four retrieval algorithms appear to be affected by clouds between altitudes of 2 and 3 km. Consequently, the Aeolus bins within this altitude range have been excluded according to the cloud-filtering method. Additionally, it is evident that in the lowest altitude bins, from 0 to 1 km, the pure-dust L2A backscatter coefficients derived from the three retrieval algorithms (SCA, SCA-MB, and MLE) – especially from SCA -- display unrealistically high values, primarily caused by surface-related effects or the increased Signal-to-Noise Ratio (SNR) levels close to the surface (Abril-Gago et al., 2022). This leads to a significant overestimation of the co-polar (L2A) backscatter coefficients and, consequently, of the retrieved L2A+ backscatter coefficients within this altitude range, with values exceeding 2 $Mm^{-1}$ $sr^{-1}$ in most retrieval algorithms. To reduce potential inaccuracies associated with surface-related effects, the comparison was limited to altitude ranges above 1 km and up to 6 km. This specific range was chosen because it encompasses the atmospheric layers where dust particles are most commonly found and where their optical and microphysical properties can be more reliably retrieved. Our findings indicate that between 1 and 6 km altitude, the Aeolus SCA, SCA-MB, MLE, and AEL-PRO co-polar (L2A) backscatter coefficients are largely underestimated throughout most of the detected dust layers. We also notice that the underestimation in the L2A backscatter signals retrieved by the SCA and MLE algorithms appears to be interrupted around 3 km altitude, where the satellite-derived L2A backscatter coefficients show a marked improvement in agreement with the eVe lidar measurements. However, the overall discrepancies between Aeolus and ground-based retrievals in the greatest part of the identified dust layers reflect the inability of the Aeolus' lidar system (ALADIN) to detect the cross-polar backscatter component of the circularly polarized emitted light when depolarizing atmospheric particles being probed, resulting in the underestimation of the backscatter coefficient (Gkikas et al., 2023).

In the comparison between the corrected pure-dust Aeolus backscatter profiles (L2A+) and the ground-based retrievals presented in Figure 13, the results indicate a substantial reduction in the Aeolus–eVe discrepancies across all retrieval algorithms. Overall, the L2A+ backscatter profiles display significantly improved agreement with the ground-based lidar measurements throughout most of the identified dust aerosol layer. A detailed, one-by-one analysis further reveals that above 3 km altitude, the SCA-MB and MLE retrieval algorithms effectively reproduce the vertical structure of the backscatter coefficient profiles retrieved by the eVe lidar, highlighting their enhanced consistency in representing the dust layer in this altitude range. In contrast, for the SCA retrieval algorithm, the reconstructed pure-dust L2A+ backscatter coefficient exhibits a slight overestimation at around 3 km altitude, followed by an underestimation between 3 and 6 km when compared with the corresponding ground-based eVe backscatter retrievals. Similarly, in the case of the AEL-PRO algorithm, the L2A+ backscatter coefficient within the 3–6 km altitude range remains underestimated relative to the ground-truth observations, although the bias is less pronounced than in the uncorrected L2A product. Furthermore, within the 1–2 km altitude interval, the pure-dust L2A+ backscatter coefficients derived from all retrieval algorithms continue to show a modest underestimation compared to the eVe lidar observations; nevertheless, the magnitude of these deviations has been considerably mitigated following the L2A+ correction procedure.

a)                                b)                                c)                                d)

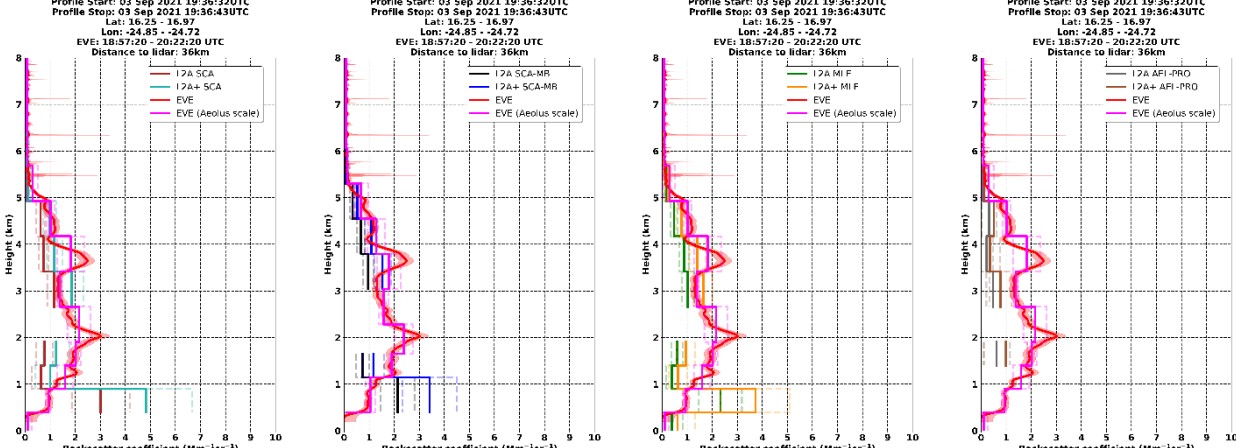

**Figure 13:** Vertical profiles of the pure-dust backscatter coefficient at 355nm retrieved by ALADIN for the L2A SCA (red), SCA-MB (black), MLE (green), and AEL-PRO (grey) products for an indicative Aeolus overpass close to Mindelo station on 3 September 2021. The vertical profiles of pure-dust backscatter coefficient at 355nm for the corrected (L2A+) SCA (cyan), SCA-MB (blue), MLE (orange), and AEL-PRO (brown) products are also illustrated. The dashed lines correspond to the estimated backscatter coefficient errors for the four retrieval algorithms. The mean pure-dust backscatter coefficient profile at 355nm retrieved from the ground-based eVe lidar on the specific date and over the time 18:57 to 20:22UTC is also provided at both the original vertical resolution of the eVe lidar (red), and the vertical resolution of the Aeolus products (magenta).

Figure 14 shows for this indicative case study, the comparison between the ground-based Polly[XT] lidar pure-dust backscatter profiles and those retrieved by Aeolus using the four algorithms: SCA (a), SCA-MB (b), MLE (c), and AEL-PRO (d). This comparison is based on nighttime observations from the Polly[XT] lidar acquired between 19:00 and 19:59 UTC, as depicted in the figure. Similar to the Aeolus-eVe profile comparison, the Polly[XT] backscatter profile is presented both at its original vertical scale (purple line) and adjusted to match the Aeolus regular and middle-bin vertical scales (pink line).

As illustrated in Figure 14, the retrieved pure-dust L2A backscatter coefficient at 355 nm profiles from all four algorithms generally underestimate the backscatter signal across most of the detected dust layers between 1 and 6 km altitude. This systematic underestimation suggests that the retrieval algorithms may not fully capture the true aerosol loading within the dust-dominated regions, due to the misdetection of the cross-polarized backscatter component. However, this underestimation in the L2A backscatter signals retrieved by the SCA, SCA-MB, and MLE algorithms appear to be interrupted around 3km altitude, where the satellite shows a slight overestimation of the L2A backscatter coefficient, particularly pronounced in the SCA, and MLE retrievals. This overestimation can be partially explained by residual cloud contamination that was not fully removed by the cloud-filtering method. Thin or fragmented clouds may still contribute to increased backscatter signals, leading to localized overestimations. Conversely, at this altitude, the pure-dust L2A backscatter values retrieved by the AEL-PRO algorithm exhibit closer consistency with the corresponding Polly[XT] lidar measurements.  The comparison between the corrected pure-dust Aeolus backscatter profiles (L2A+) and the ground-based Polly[XT] lidar retrievals reveals a notable reduction in the discrepancies between the two datasets across all retrieval algorithms. In particular, for the lofted aerosol layers between 3.5 and 6 km altitude, the adjusted L2A+ backscatter profiles—especially those retrieved using the MLE algorithm—demonstrate good agreement with the Polly[XT] observations after accounting for the missing cross-polarized backscatter component. At lower altitudes (1–2 km), although the differences between Aeolus and Polly[XT] backscatter signals also decrease across all four algorithms, the satellite retrievals continue to underestimate the observed backscatter. This residual bias can be partly attributed to the disparity in vertical resolution between Aeolus and Polly[XT]. Owing to its finer vertical resolution, Polly[XT] is capable of resolving small-scale aerosol structures and subtle variations within the dust layers, whereas Aeolus, with its coarser vertical resolution, tends to smooth these fine-scale features, leading to a dampening of localized backscatter enhancements and consequently an underestimation of the true signal.

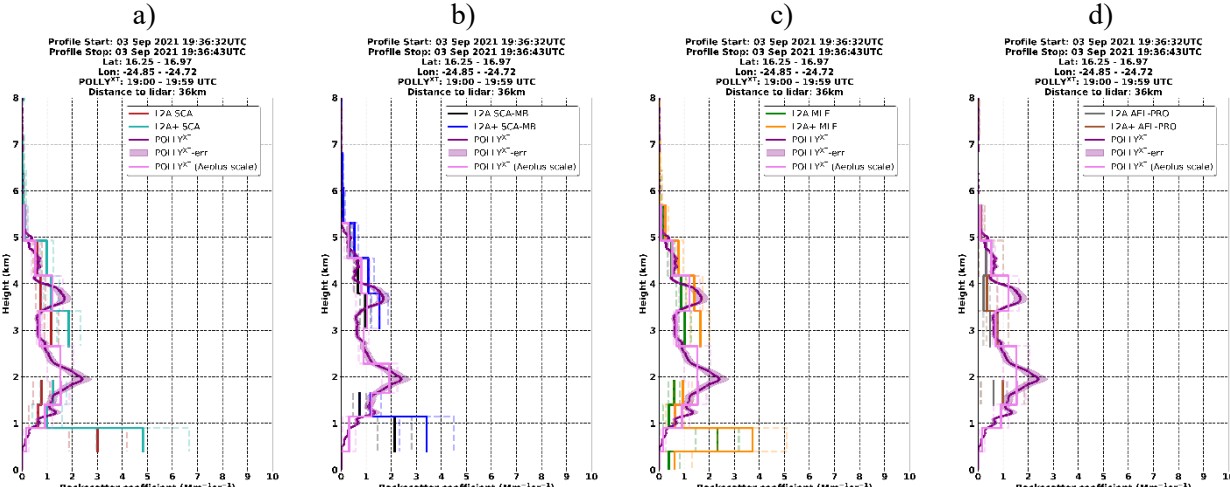

**Figure 14:** Vertical profiles of the pure-dust backscatter coefficient at 355nm retrieved by ALADIN for the L2A SCA (red), SCA-MB (black), MLE (green), and AEL-PRO (grey) products for an indicative Aeolus overpass close to Mindelo station on 3 September 2021. The vertical profiles of pure-dust backscatter coefficient at 355nm for the corrected (L2A+) SCA (cyan), SCA-MB (blue), MLE (orange), and AEL-PRO (brown) products are also illustrated. The dashed lines correspond to the estimated backscatter coefficient errors for the four rertieval algorithms. The mean pure-dust backscatter coefficient profile at 355nm acquired from the ground-based Polly[XT] lidar on the specific date and over the time 19:00 to 19:59UTC is also presented at both the original vertical resolution of the Polly[XT] lidar (purple), and the vertical resolution of the Aeolus products (pink).

Although this validation was qualitative and limited to a single case study, the results clearly indicate that incorporating the cross-polar backscatter correction substantially improves the agreement between Aeolus L2A[+] and ground-based lidar observations in dust-laden atmospheric layers. Among the evaluated retrieval algorithms, the MLE algorithm demonstrates the best overall performance for this representative case, showing strong consistency with the eVe lidar backscatter profiles across most of the identified dust layers. These findings highlight the robustness and reliability of the MLE retrieval approach in accurately capturing the vertical structure and magnitude of the dust backscatter signal as measured by ALADIN.

## 5.2 Statistical comparison

The overall assessment of Aeolus's original (L2A) and corrected (L2A+) dust backscatter coefficients was conducted using collocated backscatter observations from the ground-based eVe and Polly[XT] lidars over the entire duration of the ASKOS experiment at the Cabo Verde/Mindelo station. For the Aeolus-eVe comparison, the dataset includes collocated satellite and ground-based observations during eVe lidar's two-phase operation in Cabo Verde, specifically in July/September 2021 and June/September 2022. For the Aeolus-Polly[XT] comparison, the dataset encompasses three months of Polly[XT] operation at the Mindelo station, namely September 2021 and June/September 2022. In total, 14 collocated Aeolus-eVe profiles and 10 Aeolus-Polly[XT] profiles were collected throughout the period to validate the Aeolus pure-dust L2A and L2A+ backscatter retrievals. For the comparison, the ground-based eVe/Polly[XT] profiles were averaged within each Aeolus BRC bin to align with the vertical resolution of the L2A/L2A+ products, using both the standard (24 bins) and middle-bin (23 bins) scales.

Figure 15 displays the final dataset of Aeolus versus eVe dust backscatter coefficients over the 4-month period. The left panel shows a comparison of Aeolus pure-dust backscatter coefficients obtained from the original (L2A) products generated by the four applied algorithms. On the right, the comparison is shown for the reconstructed (L2A+) backscatter datasets. The corresponding evaluation metrics are provided in Table 2. It is important to highlight that this comparison analysis also presents evaluation metrics derived from the comparison between AEL-PRO and eVe backscatter retrievals. However, it should be noted that the AEL-PRO dataset used in this analysis has undergone significant reduction due to the cloud-filtering process and the selection of pure aerosol layers based on the AEL-PRO

classification product. These preprocessing steps were crucial for maintaining the quality and accuracy of the dataset; however, they have led to a smaller sample size for the AEL-PRO data in comparison to the datasets generated by the three other applied algorithms. Despite this reduction, the evaluation provides valuable insights into the performance and reliability of the AEL-PRO system when compared to eVe backscatter retrievals. In the comparison, all Aeolus-eVe matched bin pairs within the 1-6 km altitude range were included. This range was chosen to focus on altitudes where dust particles originating from North Africa are most concentrated, while minimizing potential contamination from surface-related effects in the calculated metrics. As shown in Fig. 15, data pairs represented by the black dashed line (y=x) indicate perfect backscatter alignment between the Aeolus-derived L2A and L2A+ products and the corresponding eVe observations. In addition, the regression linear fit offers valuable insights into the relationship between the satellite and ground-based backscatter retrievals. Specifically, the slope of the regression line reflects how closely the satellite-retrieved backscatter coefficients align with the ground-based values, while the intercept represents the associated error. Deviations from these values highlight biases in the Aeolus-retrieved backscatter coefficients. Additional information about the relationship between satellite and ground-based retrievals is provided by the Pearson correlation coefficient (R), the root mean square error (RMSE), and the mean absolute bias. Specifically, bias and RMSE metrics have been used in a complementary way to avoid any misleading interpretation of the bias score attributed to counterbalancing negative and positive satellite-lidar deviations. All the aforementioned statistical metrics have been calculated and are summarized in Table 2.

Our analysis demonstrates that the evaluation of the pure-dust Level-2A (L2A) Aeolus products revealed notable systematic biases in the retrieved backscatter coefficients. Specifically, the mean differences between Aeolus and eVe reference backscatter (calculated as *Aeolus minus eVe*) were approximately –0.98 and –0.99 Mm⁻¹sr⁻¹ for the SCA-MB and MLE algorithms, respectively, –0.91 Mm⁻¹sr⁻¹ for SCA, and –0.79 Mm⁻¹sr⁻¹ for AEL-PRO. The corresponding root mean square error (RMSE) values ranged from 1.15 Mm⁻¹sr⁻¹ for AEL-PRO to 1.39 Mm⁻¹sr⁻¹ for SCA-MB, indicating varying degrees of dispersion around the mean differences. These consistently negative biases suggest that the Aeolus L2A retrievals tend to underestimate the atmospheric backscatter signal in dust-dominated conditions. This systematic underestimation is linked to the absence of the cross-polarized component in the ALADIN lidar signal, which is particularly relevant for non-spherical aerosol particles such as mineral dust. As a result, Aeolus captures only part of the true backscattered signal, leading to a negative offset in the derived backscatter coefficients. The regression analysis between Aeolus and eVe backscatter further highlights differences in retrieval performance among the algorithms. The slope of the linear fit between Aeolus and eVe backscatter values was 0.24 for SCA, 0.16 for SCA-MB, 0.28 for MLE, and 0.27 for AEL-PRO. These low slope values reinforce the observed underestimation tendency, as they indicate that increases in eVe backscatter are only partially reflected in the corresponding Aeolus measurements. Moreover, correlation analysis yielded Pearson coefficients of approximately 0.59 for MLE and 0.55 for AEL-PRO, indicating a moderate-to-good linear relationship between Aeolus and the collocated eVe datasets. In contrast, the SCA and SCA-MB algorithms showed weaker correlations of about 0.41 and 0.28, respectively, suggesting a lower degree of consistency and potentially higher sensitivity to noise or algorithmic uncertainties in those retrieval schemes.

The comparison of the enhanced Level 2A+ (L2A+) Aeolus pure-dust backscatter coefficients against collocated eVe reference measurements reveals substantial improvements in retrieval performance across all algorithms. Notably, the incorporation of the previously missing cross-polarized component into the backscatter retrievals significantly enhances the agreement between Aeolus and eVe observations. Pearson's correlation coefficients show consistent increases for all algorithms when transitioning from the original L2A to the L2A+ product. Specifically, the correlation improves from 0.41 to 0.48 for the SCA retrievals, from 0.28 to 0.34 for the SCA-MB, from 0.59 to 0.67 for the MLE, and from 0.55 to 0.67 for the AEL-PRO algorithm. These enhancements indicate a stronger linear relationship between Aeolus and eVe datasets and reflect a notable reduction in random variability and retrieval noise in the L2A+ data. The improvement in retrieval accuracy is further confirmed by the corresponding changes in the linear regression slopes. For the SCA algorithm, the slope increases from 0.24 (L2A) to 0.30 (L2A+), while for SCA-MB it rises from 0.16 to 0.19. Similarly, the MLE algorithm exhibits an increase from 0.28 to 0.34. The AEL-PRO retrieval demonstrates the most pronounced enhancement, with a slope of 0.39, an intercept near 0.34, and a Pearson correlation coefficient of 0.67, despite the relatively limited number of available data points. These higher slope values indicate that the L2A+ retrievals more faithfully reproduce the variability observed in the eVe reference data, reducing the underestimation previously evident in the L2A results. The integration of the cross-polar component also leads to clear improvements in bias and overall error statistics. The mean absolute bias between Aeolus and eVe backscatter coefficients is reduced for all retrieval algorithms. For SCA, the absolute bias decreases from –0.91 Mm⁻¹sr⁻¹ (L2A) to –0.77 Mm⁻¹sr⁻¹ (L2A+), while similar improvements are observed for SCA-MB and MLE. The AEL-PRO retrieval,

in particular, exhibits the most significant reduction in bias, from –0.79 to –0.58 Mm⁻¹sr⁻¹, alongside a comparatively low root mean square error (RMSE) of 0.94 Mm⁻¹sr⁻¹, underscoring its enhanced accuracy. Lower RMSE values are also obtained for the MLE L2A+ product (1.17 Mm⁻¹sr⁻¹), outperforming both the SCA (1.22 Mm⁻¹sr⁻¹) and SCA-MB (1.29 Mm⁻¹sr⁻¹) retrievals.

These results highlight the significant strides made in the reliability and quality of Aeolus backscatter retrievals with the incorporation of the cross-polar component, underscoring the effectiveness of the L2A+ enhancements in addressing the previous limitations of the L2A dataset, particularly for non-spherical aerosol layers such as mineral dust.

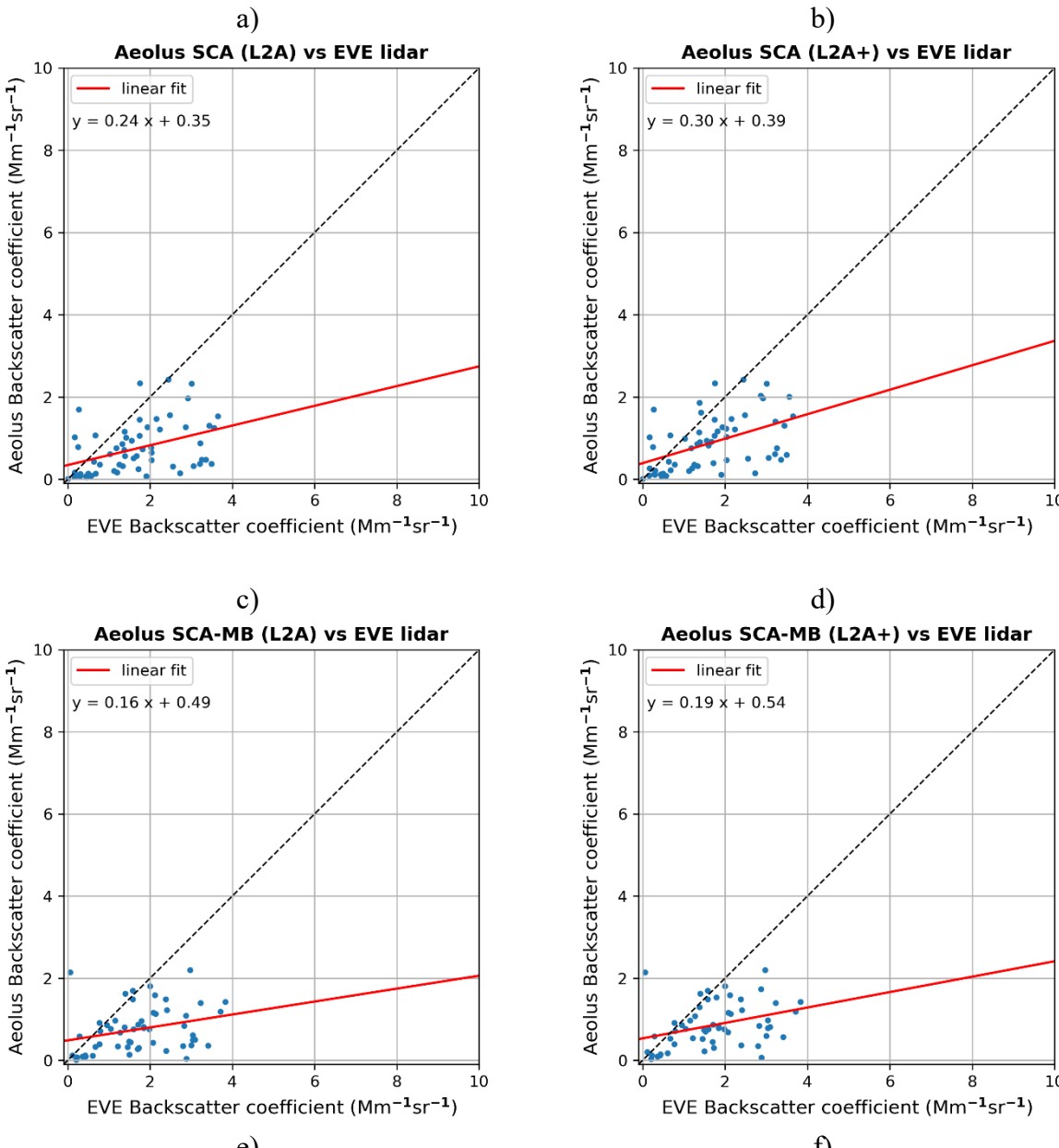

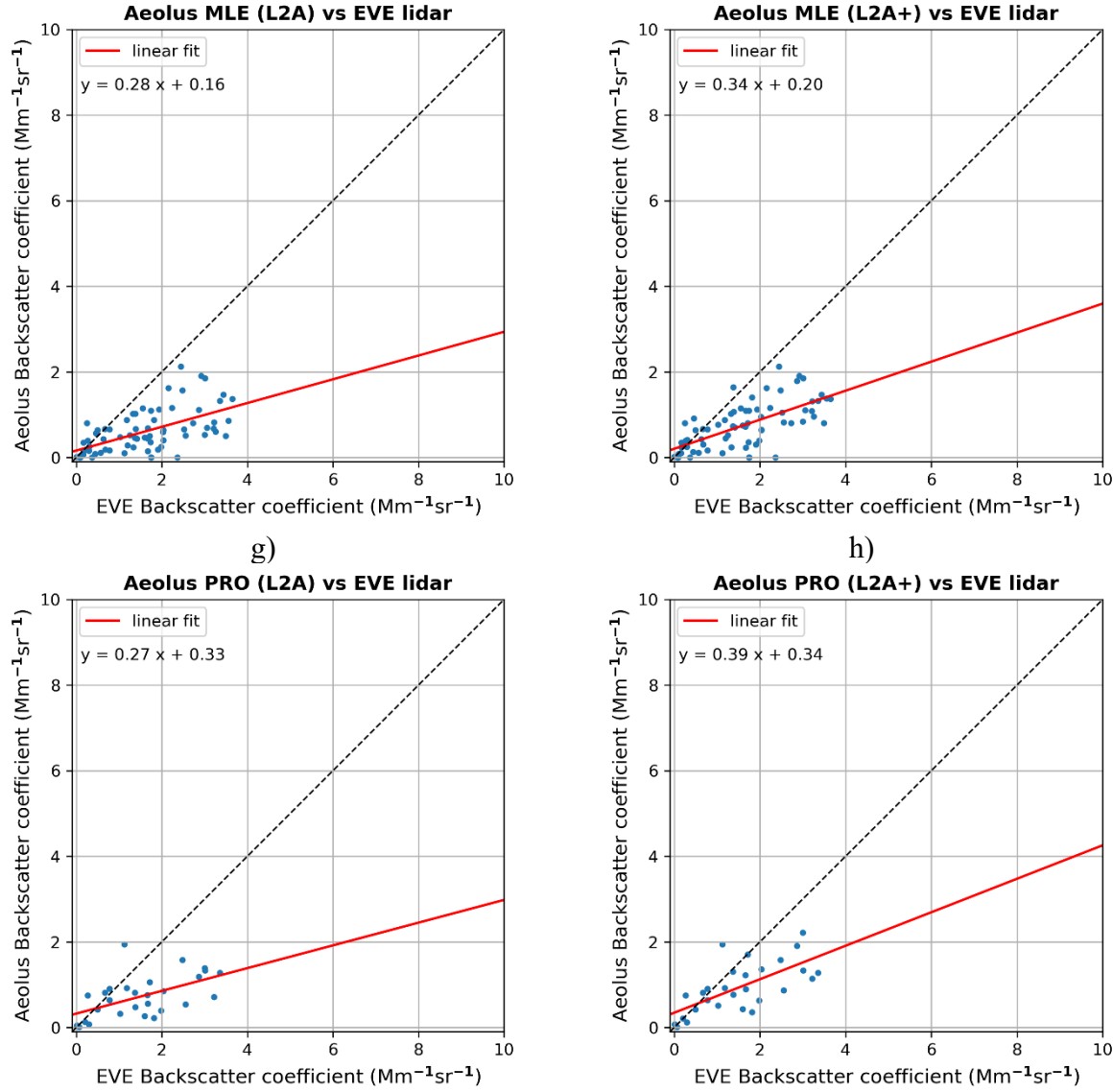

g)                                                        h)

**Figure 15:** Scatterplot comparison between Aeolus (in y axis) and eVe ground-based (in x axis) backscatter coefficient retrievals during the 4-month period of eVe/ASKOS operations at Cape Verde/Mindelo station. In the left and right panels, the results for the original (L2A) and corrected (L2A+) Aeolus retrievals are presented respectively and separately for the SCA (a, b), SCA-MB (c,d), and MLE (e,f) applied algorithms

| Algo | L2A | | | | | | L2A+ | | | | | |
| --- | --- | --- | --- | --- | --- | --- | --- | --- | --- | --- | --- | --- |
| | N | Slope | Intercept | $B_a$ | RMSE | R | N | Slope | Intercept | $B_a$ | RMSE | R |
| SCA | 59 | 0.24 | 0.35 | -0.91 | 1.34 | 0.41 | 59 | 0.3 | 0.39 | -0.77 | 1.22 | 0.48 |

| | | | | | | | | | | | |
|---|---|---|---|---|---|---|---|---|---|---|---|
| SCA-MB | 55 | 0.16 | 0.49 | -0.98 | 1.39 | 0.28 | 55 | 0.19 | 0.54 | -0.87 | 1.29 | 0.34 |
| MLE | 71 | 0.28 | 0.16 | -0.99 | 1.32 | 0.59 | 71 | 0.34 | 0.2 | -0.85 | 1.17 | 0.67 |
| PRO | 40 | 0.27 | 0.33 | -0.79 | 1.15 | 0.55 | 40 | 0.39 | 0.34 | -0.58 | 0.94 | 0.67 |

**Table 2:** Statistical indicators acquired from the comparison of the Aeolus pure-dust L2A/L2A+ backscatter retrievals (in Mm$^{-1}$sr$^{-1}$) with ground-based observations from eVe lidar during the 4-month period of ASKOS experiment at Cape Verde/Mindelo station. The following statistical parameters are included: total number of matched Aeolus-eVe pairs (N), mean absolute bias ($B_a$), root-mean-square error (RMSE), and correlation coefficient R.

Figure 16 presents scatter plot comparisons between Aeolus pure-dust L2A and enhanced L2A+ backscatter retrievals and coincident Polly$^{XT}$ ground-based lidar measurements, with separate analyses conducted for each of the four Aeolus retrieval algorithms (SCA, SCA-MB, MLE, and AEL-PRO). The corresponding statistical evaluation metrics derived from these comparisons are comprehensively summarized in Table 3.

Focusing first on the left panel of Figure 16, which depicts the comparison between the Aeolus L2A backscatter coefficients and the collocated Polly$^{XT}$ observations, it is evident that Aeolus systematically underestimates the atmospheric backscatter signal across all retrieval algorithms. This consistent underestimation pattern, similar to that observed in the Aeolus–eVe comparison, highlights the limitations of the L2A product when applied to non-spherical aerosol particles such as mineral dust. The slopes of the linear regressions between Aeolus L2A and Polly$^{XT}$ datasets
range from 0.27 to 0.31, while the corresponding Pearson correlation coefficients vary between 0.54 and 0.62, depending on the retrieval algorithm used. These relatively modest slopes and moderate correlations indicate that the L2A retrievals capture only a portion of the total backscatter measured by the ground-based lidar, primarily due to the omission of the cross-polarized signal component in the original L2A processing.

Following the implementation of the cross-polar correction in the L2A+ processing chain, the retrieval performance
shows noticeable improvement, as illustrated in the right panel of Figure 16. The regression slopes increase to 0.34 for the SCA algorithm, 0.32 for SCA-MB, and 0.30 for MLE and AEL-PRO, demonstrating a more accurate representation of the total atmospheric backscatter. This adjustment is accompanied by a general reduction in the absolute bias between Aeolus and Polly$^{XT}$ backscatter coefficients. For the SCA algorithm, the mean bias decreases from –1.18 Mm$^{-1}$sr$^{-1}$ to –1.09 Mm$^{-1}$sr$^{-1}$, while the SCA-MB and MLE algorithms exhibit reductions from –0.96 and
905 –1.23 Mm$^{-1}$sr$^{-1}$ to –0.90 and –1.12 Mm$^{-1}$sr$^{-1}$, respectively. The AEL-PRO retrieval also shows a consistent improvement, with the bias decreasing from –1.18 Mm$^{-1}$sr$^{-1}$ to –1.09 Mm$^{-1}$sr$^{-1}$.

In addition to bias reduction, the L2A+ retrievals exhibit lower root mean square error (RMSE) values, reflecting enhanced consistency with the Polly$^{XT}$ reference measurements. The RMSE for the SCA algorithm decreases from 1.61 to 1.53 Mm$^{-1}$sr$^{-1}$, for SCA-MB from 1.44 to 1.40 Mm$^{-1}$sr$^{-1}$, for MLE from 1.62 to 1.51 Mm$^{-1}$sr$^{-1}$, and for AEL-
910 PRO from 1.54 to 1.46 Mm$^{-1}$sr$^{-1}$. These reductions indicate that the L2A+ processing leads to a more stable and accurate retrieval of the atmospheric backscatter coefficient, particularly in dust-dominated scenes.

a)
b)

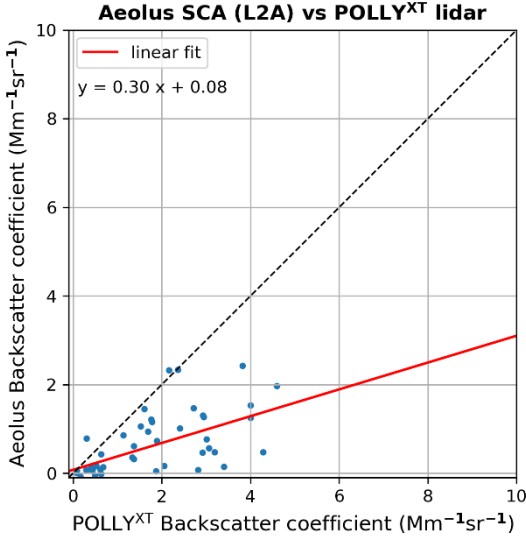

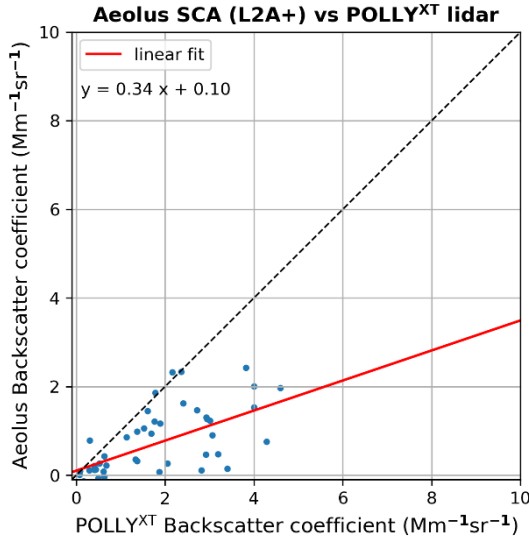

c)

d)

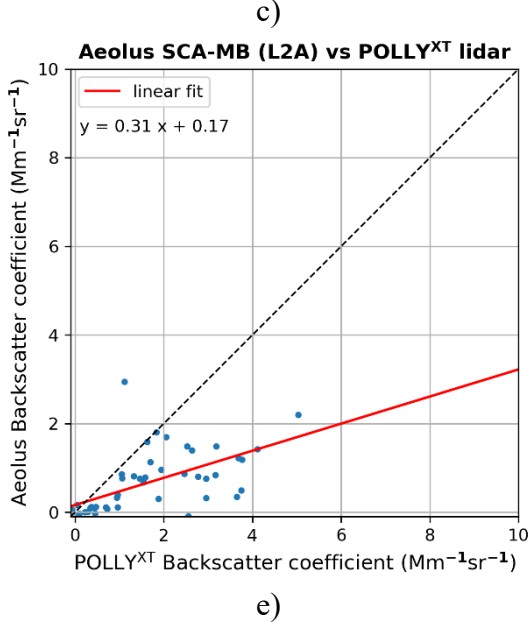

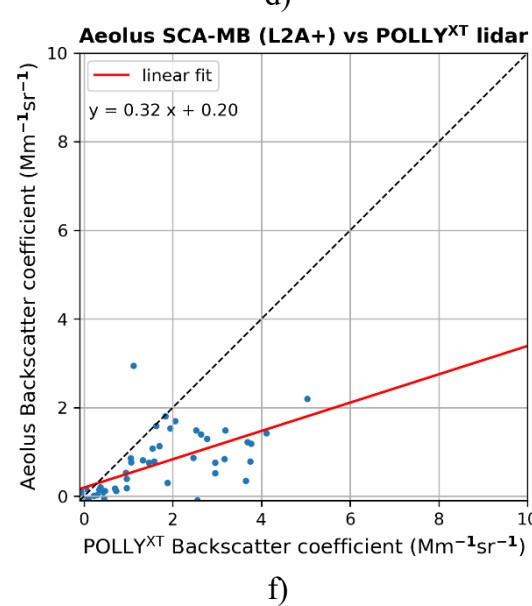

e)

f)

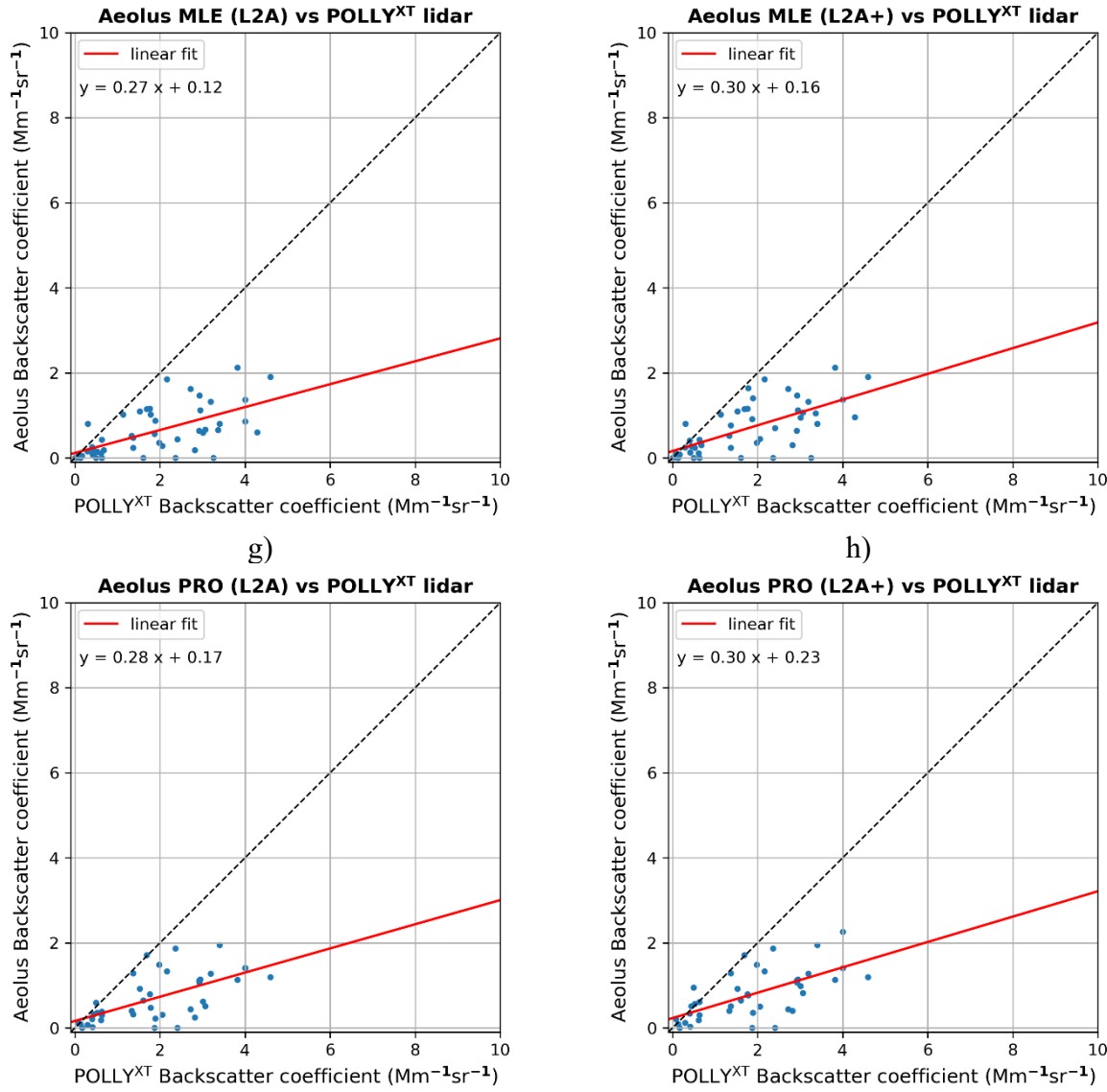

**Figure 16:** Scatterplot results as in Fig. 14 but for the comparison between the Aeolus pure-dust L2A/L2A+ backscatter retrievals with observations from Polly[XT] ground-based lidar.

| Algo | | L2A | | | | | | L2A+ | | | | |
|------|---|-------|-----------|-------|------|------|---|-------|-----------|-------|------|------|
| | N | Slope | Intercept | $B_a$ | RMSE | R | N | Slope | Intercept | $B_a$ | RMSE | R |
| SCA | 47 | 0.3 | 0.08 | -1.18 | 1.61 | 0.54 | 47 | 0.34 | 0.1 | -1.09 | 1.53 | 0.57 |
| SCA-MB | 47 | 0.31 | 0.17 | -0.96 | 1.44 | 0.58 | 47 | 0.32 | 0.2 | -0.9 | 1.4 | 0.6 |
| MLE | 48 | 0.27 | 0.12 | -1.23 | 1.62 | 0.61 | 48 | 0.3 | 0.16 | -1.12 | 1.51 | 0.66 |

| | | | | | | | | | | | |
|---|---|---|---|---|---|---|---|---|---|---|---|
| PRO | 35 | 0.28 | 0.17 | -1.18 | 1.54 | 0.62 | 35 | 0.3 | 0.23 | -1.09 | 1,46 | 0.63 |

**Table 3:** As in Table 2 but for the comparison between the Aeolus pure-dust L2A/L2A+ backscatter retrievals (in Mm[-1]sr[-1]) with ground-based observations from the Polly[XT] lidar during the ASKOS experiment.

In summary, both the Aeolus–Polly[XT] and Aeolus–eVe comparisons confirm that the incorporation of the cross-polarized backscatter component in the L2A+ processing significantly enhances the accuracy of Aeolus dust backscatter retrievals. The AEL-PRO algorithm, in particular, demonstrates the most substantial improvement, showing higher correlation with reference measurements, increased regression slope, and notably reduced bias and RMSE values. These results highlight the effectiveness of the L2A+ enhancements in correcting the systematic underestimation of the original L2A dataset and improving the representation of non-spherical aerosol layers such as mineral dust.

**6. Conclusions**

Launched on 22 August 2018, the European Space Agency's (ESA) Aeolus satellite provided horizontal line-of-sight (HLOS) wind profiles across the troposphere and into the lower stratosphere, addressing a crucial gap in the Global Observing System (GOS). Equipped with ALADIN (Atmospheric LAser Doppler INstrument), the first ultraviolet (UV) High Spectral Resolution Lidar (HSRL) Doppler lidar deployed in space, Aeolus also delivered vertically resolved optical properties of aerosols and clouds. However, one of the limitations of the ALADIN system was the absence of a cross-polarization channel that prevented accurate detection of depolarizing particles in the atmosphere, such as non-spherical aerosols and certain cloud particles. This gap impacts the retrieval of optical properties in regions where such particles are prevalent, which can hinder the precise characterization of atmospheric conditions, especially for non-spherical aerosol types like dust. More specifically, the absence of the cross-polar component also limits ALADIN's ability to differentiate between various aerosol and cloud types, and in addition impacts the quality and accuracy of its measurements in certain atmospheric conditions. To overcome these limitations, the current study focused on the development of an enhanced aerosol product for Aeolus, with a particular focus on dust. This enhanced product was designed to support aerosol data assimilation in dust transport models and improve the performance of Numerical Weather Prediction (NWP) systems by providing more accurate and reliable aerosol measurements.

For the derivation of the new product, a synergistic approach was applied, involving the utilization of spaceborne retrievals/products from multi-sensors in conjunction with reanalysis numerical outputs and reference ground-based measurements. A key aspect of this study was the validation of Aeolus's primary aerosol product (L2A) and the enhanced product (L2A+), specifically looking at profiles of particulate backscatter coefficient (baseline 16). Four different retrieval algorithms were assessed in this study: the Standard Correct Algorithm (SCA), the Standard Correct Algorithm at the middle-bin vertical scale (SCA-MB), the Maximum Likelihood Estimation (MLE), and the AEL-PRO algorithm. The performance of these algorithms was compared against corresponding ground-based measurements acquired from the eVe and PollyXT lidar systems, which were deployed during the ASKOS/JATAC experiment at the Cabo Verde/Mindelo station.

The validation and statistical analysis revealed noteworthyimprovements in the Aeolus backscatter retrievals after integrating the cross-polar component in the L2A+ product. Comparisons with ground-based lidars showed that all algorithms consistently underestimated backscatter coefficients in the original L2A product, largely due to the inability to capture the cross-polar backscatter signal. On the other hand, the L2A+ enhancements significantly reduced these discrepancies, particularly for the MLEand AEL-PROalgorithms, which achieved increased correlation coefficients, lower biases and RMSE values and better agreement with ground-based observations across most detected dust layers. For instance, AEL-PRO exhibited a remarkable increase in the correlation coefficient from 0.55 (L2A) to 0.67 (L2A+),

reduction in absolute bias, from -0.79Mm⁻¹sr⁻¹ (L2A) to -0.58Mm⁻¹sr⁻¹ (L2A+), along with the lowest RMSE of 0.94 Mm⁻¹sr⁻¹ in the Aeolus-eVe comparison. Improvements were also evident in correlation coefficients, with values increasing from 0.41 to 0.48 for SCA, from 0.28 to 0.34 for SCA-MB, and from 0.59 to 0.67 for the MLE in Aeolus-eVe comparisons, highlighting better alignment with ground truth. Similarly, in Aeolus-Polly$^{XT}$ comparisons, L2A+ adjustments notably reduced biases and improved correlation coefficients and regression slopes, particularly for the

AEL-PROMLE algorithm. More specifically, the L2A+ corrections markedly enhanced the agreement with Polly$^{XT}$ observations, increasing correlation coefficients to 0.63 for the AEL-PROalgorithm and 0.66 for the MLE algorithm, while reducing absolute biases to -1.09 and -1.12 Mm⁻¹sr⁻¹, respectively. Furthermore, the RMSE decreased from 1.54 Mm⁻¹sr⁻¹ to 1.46 Mm⁻¹sr⁻¹ for the AEL-PRO algorithm and from 1.62 Mm⁻¹sr⁻¹ to 1.51 Mm⁻¹sr⁻¹ for the MLE algorithm.

Overall, the incorporation of the cross-polar backscatter component markedly improved the Aeolus L2A product, with the AEL-PRO and MLE algorithms showing particularly strong performance gains in terms of correlation, bias reduction, and RMSE when compared to ground-based measurements. However, discrepancies between Aeolus and the ground-based lidar systems persist, largely due to Aeolus's inherently coarser vertical resolution, which smooths out the fine-scale aerosol structures and localized backscatter enhancements that are well resolved by the high-

resolution eVe and PollyXT lidars. These differences are further influenced by the limited number of available collocated measurements, a consequence of both the cloud-filtering implementation applied to Aeolus data and the overall small sample size of coincident observations. Additionally, the coarse horizontal resolution of the BRC-level Aeolus L2A products (~90 km) limits the instrument's ability to resolve small-scale aerosol gradients in the lower troposphere, which also justified our choice to retain only the nearest eVe/Polly$^{XT}$–Aeolus collocations per overpass.

The potential benefits of finer-resolution Aeolus profiles (e.g., ~18 km from MLEsub products) are discussed in Trapon et al. (2025). Despite these limitations, the enhanced L2A+ product—especially when processed with AEL-PRO and MLE—represents a significant step forward, offering more accurate dust aerosol retrievals that improve the initialization of dust transport models, support assessments of aerosol–cloud–radiation interactions, and contribute to more reliable Numerical Weather Prediction (NWP) and climate applications.


**Author contribution**

KR: Data curation, Formal analysis, Methodology, Writing – original draft preparation. EP: Conceptualization, Methodology, Supervision, Writing – review & editing. TG: Resources, Software, Writing – review & editing. AG: Conceptualization, Supervision, Writing – review & editing. EM: Resources, Supervision, Writing – review & editing. PP: Resources, Writing – review & editing, KAV: Resources, Writing – review & editing. AT: Writing – review & editing. DD: Resources, Writing – review & editing, GZ: Resources. AB: Resources. HB: Resources. AAF: Resources, Writing – review & editing. Nikos Benas: Resources, Writing – review & editing, MS: Resources. CR: Supervision.

EM: Supervision. VA: Conceptualization, Project administration, Supervision.

**Competing interests**

Some authors are members of the editorial board of AMT.


**Acknowledgements**

We are grateful to ESA for making available the AEOLUS products used in this study. We gratefully acknowledge the use of the CAMS (Copernicus Atmosphere Monitoring Service) reanalysis dataset, made available by the European Centre for Medium-Range Weather Forecasts (ECMWF). We sincerely appreciate the efforts of the CAMS team in maintaining and providing access to this valuable dataset for scientific research.


**Financial support**

This research study has been funded by the European Space Agency under the L2A+ project (contract no. 4000139424/22/I-NS) and the AIRSENSE project (contract no. 4000142902/23/I-NS), the PANGEA4CalVal project (Grant Agreement 101079201), funded by the European Union, the AXA Research Fund (Earth Observation for Air-Quality - Dust Fine-Mode (EO4AQ-DustFM), Hellenic Foundation for Research and Innovation (H.F.R.I.) under the "3rd Call for H.F.R.I. Research Projects to support Post-Doctoral Researchers" (Project Acronym: REVEAL, Project Number: 07222). K.R. also acknowledges support from the H.F.R.I. under the '2nd Call for H.F.R.I. Research Projects to Support Post-Doctoral Researchers' (Project StratoFIRE, Project Number 3995)."

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
