# Peer review of "Derivation and validation of a refined dust product from Aeolus (L2A+)"

_EGUsphere, 2025_

## Referee Comment (RC2)

**A Review of "Derivation and validation of a refined dust product from Aeolus (L2A+)" by Konstantinos Rizos et al., Atmos. Meas. Tech. Discuss., https://doi.org/10.5194/egusphere-2025-1175, 2025**

Dimitri Trapon (dimitri.trapon@tropos.de)

The paper is well structured and provides valuable insights into the potential of Aeolus aerosol data product Level-2A labelled Baseline 16 for the characterization of dust particles. Kindly consider the comments attached which may help clarifying some points.

General comments:

1)  The study makes use of the extinction and backscatter coefficients for particles retrieved by the Aeolus Level-2A AEL_PRO algorithm which is initially provided at measurement level ~ 3 km horizontal, and averaged onto the coarser resolution BRC ~ 90 km to be compared with BRC-level retrievals from SCA and MLE algorithms. But the principle of AEL_PRO profiling using optimal estimation and features identification may induce deviation in their horizontal neighbooring (Wang et al., 2024). Therefore, any variability happening in the consecutive 30 measurements of AEL_PRO are not considered here right ? Did the authors take a look at some heteroegenity indicators per BRC ?.Are the statistics impacted when using 1st, 15th or last AEL_PRO measurement profile instead ? And what if a localized cloud is detected , e.g. by the 20th measurement : is the entire BRC flagged as cloud contaminated condition, or is the averaging still perfomed between 1st and 19th measurement ? Kindly clarify.

2)  The study makes use of a cloud screening method derived from Aeolus Level-2A AEL_FM and CLAAS-3/SEVIRI. It is described as a complex method, which is not straightforward because of the measurement level ~ 3 km of AEL_FM compared to coarser horizontal BRC ~ 90 km for SCA and MLE. The L2A product labelled Baseline 16 includes a ready-to-use cloud mask for SCA and MLE (ATBD, Flamant et al., 2022), given at BRC level and based on ECMWF model, i.e. using cloud liquid water content and cloud ice water content and implemented by L2A processor version 3.13. Can the authors clarify why they have not selected it ? If some tests have been performed, a short paragraph describing the outcomes may be added to the text.

3)  Aeolus only measure the co-polarized backattered light, therefore underestimating the total backscatter for highly depolarizing particles such as Saharan dust. In the text an underestimation factor of ~ 33 % is mentionned for pure dust (line 128). But when looking at Fig. 10 the L2A+ total backscatter do not seem significantly increased accordingly, even after removal of cloud contaminated bins and dust-free bins. Could the authors discuss further this point ? Is it because of a lower computed circular depolarization ratio than expected for pure dust ? Is the linear particle depolarization ratio value of 0.244 applied for Saharan dust with Eq. 3 in cause ? The selected continuous linear colormap may also play a role. Kindly consider adjustment of colormap to better highlight the increase, or adding difference map in appendix.

4)  The study relies on both the circular depolarization ratio at 355 nm derived with Eq.3 from the data collection DeLiAn, and aerosol typing method based on CAMS reanalysis products. The depolarization information then appears crucial to derive the L2A+ total backscatter, and to confirm the occurrence of irregularly shaped particles such as Saharan dust. But no tables or values of both the circular and linear depolarization ratios at 355

nm can be found in the text. Can the authors share some estimations, at least for the pure dust case of 3rd September 2021 which is illustrated in Figs. 9 - 10 ? Moreover, showing any collocated measurement of volume circular an linear depolarization ratios with Aeolus direct overpasses would be valuable as ground-truth . Are such profiling available from the instruments Polly[XT] and EVE lidar ? If yes, kindly consider discussing it, or possibly showing the profiles on 3rd September 2021. Highlighting which Aeolus profile is the direct overpass above the instruments, i.e. above Mindelo, Cabo Verde would also be a plus for Figs. 10-11.

5)   The L2A products labelled Baseline 16 include Quality Checks (QC) flags for SCA, SCA-MB, and MLE (ATBD, Flamant et al., 2022, and User Guide, Trapon et al., 2022). These flags are of high relevance, especially for noisy signals derived with the SCA in low Signal-to-Noise regions and for MLE in low altitudes below 2 km. Could the authors clarify if these quality flags have been applied to the signals ? If not applied, is it because bins removal and the will to keep as many valid bins as possible before cloud screening ? A few lines discussing this point can be added to the text, at least saying that ready-to-use QC flags are included in the L2A product.

6)   The L2A products labelled Baseline 16 include newly implemented MLE processed at finer horizontal resolution sub-BRC (MLEsub) which was implemented in L2A processor version 3.16. Each sub profile corresponds to 6 accumulated measurements with September 2021 settings. A BRC is then filled with 5 MLEsub sub-profiles .If considering the comments 1), it would be interesting to reproduce the study looking at MLEsub versus AEL_PRO ; the averaging for AEL_PRO being then less prominent and possibly impacting the cloud screening. Kindly consider mentioning this open point into the *conclusion* section, or eventually within a *code and data availability* section.

Specific comments:

Line 104 *"...through the independent estimation of the the volume extinction and co-polarized volume backscatter coefficient..."*. This is because the ability to seperate the contributions from molecules and particles with the two channels. It can be mentionned here as well (it is mentionned later in the paper by section 2.1 line 188).

Line 106 *"…from two different spectral channels (Ehlers et al., 2022)…"*. Kindly consider (Flamant et al., 2008) instead.

Line 116 *"…reprocessed in Baseline 10 (B10)…"*. Kindly consider rephrasing, i.e. *L2A processor version 3.10, referred as Baseline 10 (B10)* .

Line 124 *"…underestimation of the order of 18% of the Aeolus-like backscatter…"*. What does *Aeolus-like backscatter* mean ? Do the authors refer to the underestimation of the total backscatter ? Kindly clarify.

Line 187 *"…cross-talk coefficients…"*. The principle of the *cross-talk* correction, i.e. seperation between lidar signal contributions from particles and molecules for vertical matching between Rayleigh an Mie channels, can be mentioned here.

Line 188 *"…and molecular and particulate contributions to the signals…"*. The reference papers (Flament et al., 2020 ; Flament et al., 2021) can be added.

Line 507 *"…To address this issue…"*. Would *limitation* be more suited than *issue* ?

Line 619 *"…L2A backscatter coefficients from the three retrieval algorithms exhibit unreasonably high values, primarily due to surface-related effects (Abril-Gago et al., 2022)"*. The statement can be complemented with *", and positive bias by optimal estimation based MLE in low altitude regions below 2 km."*. A possible reference for Level-2A product quality assessment from 4th reprocessing effort and labelled Baseline 16 is available at *https://earth.esa.int/eogateway/documents/d/earth-online/aeolus-summary-reprocessing-4-fm-b-disc-2024-04-30*.

Figures 9-10-11 If considering high dust fraction up to ~ 70 % between ~ 10 km altitude and ~ 16 km altitude between profiles [-21.87°E, 0.38°N] and [-25.28°E, 18.62°N] on case 3rd September 2021 in Fig. 9b, why do we see only white background for the corresponding regions in SCA and MLE retrievals in Figs 10-11 ? Does that mean fully cloud contaminated bins which were then removed, or invalid SCA and MLE ? But SCA and MLE products do not seem too much attenuated below this regions, then between ground and ~ 10 km...The flagging method including dust-free removal and cloudy contaminated bins is difficult to apprehend for such regions. Are there any other quality flagging applied here ? Is it because unrealistic dust fraction by CAMS ? Kindly clarify.

Figure 11 Kindly consider changing the colormap to be able to visualy compare the low dust mass concentration (i.e. below 150 $\mu$g/m³) from the suroundings.

Figure 11 The DEM intersection is visible for pannels Fig. 11a-c-e-g but not for pannels Fig. 11b-d-f-h. This may be linked to a linewidth adjustment. Kindly consider re-generation of the figures.

Figures 12-13 The vertical profiles and errors in dashed lines are hardly dinstinguishable. Kindly consider 2x2 pannels with increased size instead, and possibly reducing the xaxis top limit of backscatter coefficient. Moreover, kindly consider other color combination than red-green to get colorblind-safe color scheme.

Figures 14-15 Kindly consider adjustment of yaxis and xaxis to lower values (e.g. 6 or 8 $Mm^{-1}sr^{-1}$) for better readability.

---

## Author Comment (AC1)

**Response to Reviewers**

For the article entitled "**Derivation and validation of a refined dust product from Aeolus (L2A+)**"

Konstantinos Rizos, Emmanouil Proestakis, Thanasis Georgiou, Antonis Gkikas, Eleni Marinou, Peristera Paschou, Kalliopi Artemis Voudouri, Athanasios Tsikerdekis, David P. Donovan, Gerd-Jan van Zadelhoff, Angela Benedetti, Holger Baars, Athena Augusta Floutsi, Nikos Benas, Martin Stenge, Christian Retscher, Edward Malina, Vassilis Amiridis

We would like to express our sincere gratitude to all the reviewers for their careful evaluation of our manuscript and for their insightful and detailed suggestions, which have significantly contributed to enhancing its overall quality. All recommendations have been thoroughly considered and duly incorporated into the revised version.

In the following pages, we provide the reviewers' comments (in blue, bold-italic), together with our detailed, point-by-point responses and a description of the corresponding revisions made to the manuscript.

**Reviewer: 1**

*Comments to the author:*

1. *Abstract: It could be improved to convey the key points more clearly and concisely. By reading the abstract (and the conclusions), it is unclear how the dust product is refined and improved. Lines 26–28 state: "The enhanced aerosol product is derived through a series of intermediate processing steps that integrate spaceborne retrievals/products from multiple sensors, reanalysis numerical outputs, and reference ground-based measurements." This could be rewritten more directly, for example, as: "The enhanced dust aerosol product is obtained by estimating the missing cross-polarized component using a fixed depolarization ratio for dust as a constraint. Collocated satellite measurements assist with cloud screening and aerosol subtyping." Another example, lines 28-31 state "Both the primary (L2A), and enhanced (L2A+) Aeolus optical products, in terms of profiles of backscatter coefficient at 355 nm are retrieved using four different algorithms, the Standard Correct Algorithm (SCA), the Standard Correct Algorithm at the middle-bin vertical scale (SCA-MB), the Maximum Likelihood Estimation (MLE), and AEL-PRO.", which could be rewritten as: "Both the primary (L2A) and enhanced (L2A+) Aeolus backscatter coefficient profiles at 355 nm are retrieved using four different algorithms: the Standard Correct Algorithm (SCA), the Standard Correct Algorithm with middle-bin vertical scaling (SCA-MB), the Maximum Likelihood Estimation (MLE), and AEL-PRO." The abstract begins with "The missing cross-channel…", which may give the impression that the absence of cross-polarization measurements in the Aeolus lidar resulted from a malfunction. It would be helpful to clarify—either in the abstract or in the introduction—that the cross-polarization channel was intentionally omitted by design due to the coaxial configuration of the Aeolus lidar, which was optimized primarily for Doppler wind measurements.*

**Response:**

We thank the reviewer for the constructive comments on the clarity and presentation of the abstract. All the points raised have been carefully considered and incorporated into a revised version of the abstract.

- Regarding the description of how the enhanced dust product is refined, the text has been rewritten to more clearly convey the methodology. "The L2A+ product is generated through a series of processing steps that integrate multi-sensor satellite retrievals for cloud screening and aerosol layer characterization, CAMS reanalysis outputs to classify aerosol types, distinguish dust from non-dust fractions, and provide the missing depolarization ratio values required for the Polarization Lidar

Photometer Networking (POLIPHON) technique, along with ground-based lidar measurements used for performance assessment." This revision directly addresses the reviewer's concern about clarifying how the dust product is improved.

- The sentence detailing the retrieval algorithms has been rephrased for clarity and conciseness. It now reads: "Both the primary (L2A) and enhanced (L2A+) Aeolus pure-dust backscatter coefficient profiles at 355 nm are retrieved using four different algorithms: the Standard Correct Algorithm (SCA), the Standard Correct Algorithm with middle-bin vertical scaling (SCA-MB), the Maximum Likelihood Estimation (MLE), and AEL-PRO."

- To avoid any misunderstanding, we have clarified in the beginning of the abstract that the absence of cross-polarization measurements in ALADIN is that it was originally designed as a wind lidar without depolarization measurement capability. Therefore, it detects only the co-polar component of the backscattered signal, limiting the accuracy of optical products in the presence of depolarizing atmospheric layers.

We believe that the above-mentioned revisions address the reviewer's concerns and improve the clarity, accuracy, and readability of the abstract.

2. *Equation 1: The term CPM10 should be clearly defined. Below Eq. 1, the text states: "In the above formula (Eq. 1), the sum of … multiplied by the dry air concentration inside the parenthesis ($pm/Rspec * T$) calculates the mass concentration …" This suggests that CPM10 represents the mass concentration in μg/m³, and the term in parentheses is the dry air concentration. However, it is unclear what exactly is meant by "dry air concentration" here. Based on the units provided in the paper, the expression inside the parentheses appears to have units of kg2/(J·m³), which is inconsistent. Please double-check the dimensional consistency of this term and clearly define what is meant by "dry air concentration." Additionally, the terms SS, DD, OM, etc., seem to refer to different aerosol types. However, it is unclear what specific parameter of each aerosol type is being represented (e.g., mixing ratio, mass fraction, volume concentration). To improve clarity, consider using a single character (e.g., f for fraction, c for concentration or X for mixing ratio) to denote the parameter and use the aerosol type as a subscript (e.g., XSS, XOM, etc). This would make the equation structure more interpretable and scientifically consistent.*

**Response:**

We thank the reviewer for this valuable comment and for highlighting the need for greater clarity and dimensional consistency in Equation 1. The section has been revised accordingly.

$$C_{PM10} = \left( \frac{X_{SS1} + X_{SS2} + X_{SS3}}{4.3} + X_{DD1} + X_{DD2} + X_{DD3} + X_{OM} + X_{BC} + X_{SU} \right) * \left( \frac{p_m}{R_{spec} * T} \right) \quad (1)$$

The variable $C_{pm10}$ is now explicitly defined as the total aerosol mass concentration (μg/m³), while $X_p$ denotes the mass mixing ratio (kg/kg) of aerosol type p ($SS_{1-3}$ for sea salt, $DD_{1-3}$ for dust, OM for organic matter, BC for black carbon, and SU for sulfate). The term inside the parentheses, ($p_m$ / ($R_{spec} \times T$)), has been clarified to represent the air density (kg/m³), derived from the ideal gas law. Multiplying the aerosol mass mixing ratio by this term yields the mass concentration in kg/m³, which is then converted to μg/m³. This correction ensures full dimensional consistency and provides a physically sound description of the conversion process. The notation has also been standardized using **$X_p$** to denote the mixing ratio of each aerosol type, in accordance with the reviewer's suggestion.

3.  *Section 3.3 Aerosol typing method: It appears that the Aeolus dust layer is identified based on CAMS reanalysis products, with predefined thresholds of 1.3 μg/m³ for dust mass concentration and 50% for dust fraction. It seems that the dust fraction is derived using Eq. (1); if so, please clearly explain how this equation is applied in the analysis. Regarding the dust concentration, it is unclear whether it is based solely on CAMS reanalysis data, Aeolus measurements, or a combination of both. Please clarify the data source and methodology used. Additionally, in Line 491, the phrase "the median value of the entire dust mass concentration" is ambiguous. What does "entire" refer to in this context? Does it mean all BRC bins, the full vertical profile, or something else? Please specify for clarity.*

**Response:**

We thank the reviewer for this constructive and detailed comment. In the revised version of the manuscript, Section 3.3 (*"Quantifying the Aeolus pure-dust component through reconstruction of its missing cross-polar term"*) has been substantially rewritten to clarify the aerosol typing methodology and to replace the previous threshold-based approach.

In the updated framework, Aeolus dust identification is no longer based on fixed concentration or fraction thresholds (e.g., 1.3 μg/m³ or 50%), but instead relies on a physically consistent, POLIPHON-based methodology. The one-step POLIPHON technique (Pappalardo et al., 2014) allows the decomposition of the total aerosol load into pure-dust and non-dust components by exploiting particle depolarization ratios. Since Aeolus ALADIN does not provide depolarization measurements, CAMS reanalysis products were integrated into the L2A+ workflow to (i) classify aerosols along the Aeolus overpasses, (ii) distinguish dust

from non-dust fractions, and (iii) supply the missing depolarization ratio values needed to apply the POLIPHON method.

A detailed spatiotemporal collocation between Aeolus and CAMS was performed using nearest-neighbor matching, with CAMS outputs selected within ±3 hours of each Aeolus observation. For each Aeolus BRC bin, CAMS aerosol mass concentration profiles were averaged over the corresponding vertical layer. These CAMS-derived parameters, dust mass concentration and dust fraction, were used to qualitatively identify the atmospheric layers dominated by dust. Over these layers, the missing Aeolus cross-polar backscatter component was reconstructed using Eq. (4):

$$\beta_{\text{total}}^{\text{cross}} = \delta_{circ}^{355} \times \beta_{\text{total}}^{co}$$

where $\beta_{\text{total}}^{\text{cross}}$ is the missing cross-polar contribution from dust and non-dust aerosols, and $\beta_{\text{total}}^{co}$ is the directly observed Aeolus co-polar backscatter. The particle depolarization ratio $\delta_{circ}^{355}$ is derived from CAMS-based linear depolarization ratios through the POLIPHON relation (Eqs. 2–3), incorporating theoretical values of $\delta_{355}$,d = 0.244 and $\delta_{355}$,nd = 0.03 (converted to circular values 0.65 and 0.06, respectively).

This new formulation allows the reconstruction of Aeolus's missing cross-polar term, enabling the retrieval of complete (L2A+) pure-dust and total backscatter coefficients. It also ensures that the aerosol typing is based on physically meaningful parameters, namely, particle depolarization ratio and backscatter properties, rather than on empirically defined thresholds.

Finally, to clarify the reviewer's concern, the "median value of the entire dust mass concentration" refered explicitly to the median over all Aeolus BRC bins within the identified dust layers, providing a consistent measure of dust intensity across the profile.

4. *Section 3.4 Adjustment of the missing Aeolus cross-polar backscatter component: Equations (2) and (3) are valid for pure dust. However, since a layer is classified as dust when the CAMS dust fraction exceeds 50%, it may not consist entirely of dust. Could the dust fraction be incorporated into the retrieval to account for the presence of other aerosol types? It would also be valuable to present lidar ratios derived from the L1A+ backscatter coefficient and HSRL extinction coefficient, allowing comparison with previously reported values. A related question—perhaps beyond the scope of this paper—is whether the lidar ratio could also be used as an additional constraint for identifying pure dust, alongside the depolarization ratio?*

**Response:**

In the revised version of the manuscript, Section 3.3 (*"Quantifying the Aeolus pure-dust component through reconstruction of its missing cross-polar term"*) has been expanded to clarify this point.

To produce the L2A+ dust product, the reconstruction of the missing Aeolus cross-polar component is based primarily on CAMS-derived depolarization ratio values, rather than directly on the dust fraction. Specifically, CAMS data are used to estimate the particle depolarization ratio through the one-step POLIPHON methodology, which enables the separation of dust and non-dust components in a physically consistent way. This approach allows the derived depolarization ratio to implicitly reflect the degree of dust dominance within each layer, without applying an explicit dust-fraction weighting. Therefore, even in cases where the CAMS dust fraction is below 100 %, the retrieved depolarization ratio, and consequently the reconstructed Aeolus cross-polar term, accounts for the partial presence of non-dust aerosols within the mixture.

Regarding the reviewer's suggestion on using lidar ratios, we fully agree that this parameter provides valuable information for aerosol typing. Although the present study focuses on reconstructing the Aeolus cross-polar signal and evaluating the improvement in backscatter retrievals, the extension of this framework to include lidar-ratio-based analysis and comparison with reference values from EARLINET datasets is foreseen as part of our future work.

Overall, the revised methodology ensures that mixed aerosol conditions are treated consistently through the CAMS-based depolarization ratio retrieval, while establishing a robust foundation for future inclusion of additional constraints such as the lidar ratio.

5. *Figures: The quality of the figures should be improved. Figures 12 and 13, in particular, are overly busy, making it difficult to distinguish individual curves. As a result, it is challenging to assess the agreement between profiles by visual inspection alone. To improve clarity, consider using error bars to represent uncertainties instead of plotting two separate error curves for each backscatter profile.*

**Response:**

We appreciate the reviewer's suggestion. The figures have been revised using thinner, less bold lines for the uncertainty ranges, improving clarity and allowing individual curves to be more easily distinguished.

==Reviewer: 2==

*General comments to the author:*

1. *The study makes use of the extinction and backscatter coefficients for particles retrieved by the Aeolus Level-2A AEL_PRO algorithm which is initially provided at measurement level ~ 3 km*

*horizontal, and averaged onto the coarser resolution BRC ~ 90 km to be compared with BRC-level retrievals from SCA and MLE algorithms. But the principle of AEL_PRO profiling using optimal estimation and features identification may induce deviation in their horizontal neighbooring (Wang et al., 2024). Therefore, any variability happening in the consecutive 30 measurements of AEL_PRO are not considered here right ? Did the authors take a look at some heteroegenity indicators per BRC ?.Are the statistics impacted when using 1st, 15th or last AEL_PRO measurement profile instead ? And what if a localized cloud is detected , e.g. by the 20th measurement : is the entire BRC flagged as cloud contaminated condition, or is the averaging still perfomed between 1st and 19th measurement ? Kindly clarify.*

**Response:**

We thank the reviewer for this comment. When averaging AEL-PRO measurements per BRC profile, a strict cloud-filtering procedure was applied. Specifically, any BRC containing at least one cloud-contaminated measurement, as identified by both AEL-FM and SEVIRI (MSG), was entirely excluded from the averaging. Furthermore, for the AEL-PRO algorithm, the retrieved cloud-free co-polar backscatter profiles were additionally filtered using the AEL-PRO classification product, retaining only BRC bins classified as tropospheric aerosols (index = 103). As a result, the averaged BRC profiles include exclusively cloud-free aerosol measurements, ensuring that localized clouds or contaminated features do not affect the derived statistics.

2. *The study makes use of a cloud screening method derived from Aeolus Level-2A AEL_FM and CLAAS-3/SEVIRI. It is described as a complex method, which is not straightforward because of the measurement level ~ 3 km of AEL_FM compared to coarser horizontal BRC ~ 90 km for SCA and MLE. The L2A product labelled Baseline 16 includes a ready-to-use cloud mask for SCA and MLE (ATBD, Flamant et al., 2022), given at BRC level and based on ECMWF model, i.e. using cloud liquid water content and cloud ice water content and implemented by L2A processor version 3.13. Can the authors clarify why they have not selected it ? If some tests have been performed, a short paragraph describing the outcomes may be added to the text.*

**Response:**

In our analysis, we applied a combined cloud-screening approach using both the Aeolus Feature Mask (AEL-FM) and the SEVIRI CLAAS-3 cloud-mask product to ensure more robust identification and removal

of cloud-contaminated profiles. Although the L2A Baseline 16 processor includes a cloud mask at the BRC level, we chose this combined approach to benefit from the higher horizontal resolution (~3 km) of the AEL-FM product and the complementary cloud information from SEVIRI (~4 km, 15 min temporal resolution).

The AEL-FM algorithm, included in the Baseline 16 processor used in this study, provides a feature detection probability index capable of distinguishing between clear-sky, optically thick clouds, and fully attenuated signals, allowing finer discrimination of small-scale cloud structures. The SEVIRI CLAAS-3 binary cloud mask, on the other hand, offers continuous and independent geostationary cloud observations, enabling cross-verification of cloudy scenes detected by Aeolus.

By combining these two datasets, we ensured a more conservative and reliable cloud screening, minimizing the risk of residual cloud contamination in the Aeolus optical profiles used for aerosol analysis.

3. *Aeolus only measures the co-polarized backattered light, therefore underestimating the total backscatter for highly depolarizing particles such as Saharan dust. In the text an underestimation factor of ~ 33 % is mentionned for pure dust (line 128). But when looking at Fig. 10 the L2A+ total backscatter do not seem significantly increased accordingly, even after removal of cloud contaminated bins and dust-free bins. Could the authors discuss further this point ? Is it because of a lower computed circular depolarization ratio than expected for pure dust ? Is the linear particle depolarization ratio value of 0.244 applied for Saharan dust with Eq. 3 in cause ? The selected continuous linear colormap may also play a role. Kindly consider adjustment of colormap to better highlight the increase, or adding difference map in appendix.*

**Response:**

In the earlier version of the manuscript, the identification of dust layers relied on CAMS dust mass concentration (median threshold: 1.3 µg/m³) and dust fraction (50%) to classify Aeolus BRC bins as dust-dominated. However, these thresholds do not necessarily correspond to fully pure-dust layers, as the classified bins may still contain a mixture of dust and non-dust aerosols. This partially explains why the increase in the L2A+ total backscatter in Figure 10 did not appear as strong as the theoretical ~33% underestimation factor expected for purely depolarizing dust.

In the revised version of the study, we implemented an improved, physics-based aerosol typing scheme using the one-step POLIPHON methodology. Through this approach, CAMS reanalysis data were used to estimate the particle depolarization ratio, enabling the physical separation of dust and non-dust components in the aerosol mixture. This method inherently reflects the degree of dust dominance within each layer, without requiring an explicit dust-fraction weighting. As a result, even for partially mixed layers, the retrieved depolarization ratio, and therefore the reconstructed Aeolus cross-polar contribution, appropriately accounts for the presence of non-dust aerosols, leading to a more realistic enhancement of the total backscatter signal.

Additionally, following the reviewer's suggestion, we have modified the color scale in Figure 11 (previously Figure 10) to a non-linear scheme, which enhances visual contrast and makes the increase in total backscatter after the L2A+ correction more clearly visible.

4. *The study relies on both the circular depolarization ratio at 355 nm derived with Eq.3 from the data collection DeLiAn, and aerosol typing method based on CAMS reanalysis products. The depolarization information then appears crucial to derive the L2A+ total backscatter, and to confirm the occurrence of irregularly shaped particles such as Saharan dust. But no tables or values of both the circular and linear depolarization ratios at 355 nm can be found in the text. Can the authors share some estimations, at least for the pure dust case of 3rd September 2021 which is illustrated in Figs. 9 - 10 ? Moreover, showing any collocated measurement of volume circular an linear depolarization ratios with Aeolus direct overpasses would be valuable as ground-truth . Are such profiling available from the instruments PollyXT and EVE lidar ? If yes, kindly consider discussing it, or possibly showing the profiles on 3rd September 2021. Highlighting which Aeolus profile is the direct overpass above the instruments, i.e. above Mindelo, Cabo Verde would also be a plus for Figs. 10-11.*

**Response:**

In response to the reviewer's comment, we would like to clarify that in the initial version of our methodology, the enhanced L2A+ dust product was derived by estimating the missing cross-polarized backscatter component using a fixed depolarization ratio value for dust as a constraint. For this reason, explicit tables or varying values of the circular and linear depolarization ratios at 355 nm were not included in the manuscript.

Specifically, we adopted a fixed particle linear depolarization ratio ($\delta_{355}$,d = 0.244) for Saharan dust, which was then converted to its circular equivalent using Eq. (3). This value was taken from the DeLiAn database (Floutsi et al., 2022) and was considered appropriate for our study domain over the Atlantic Ocean, where Saharan dust is the dominant aerosol type.

However, as correctly noted by the reviewer, this fixed-value approach limits applicability to regions dominated by other aerosol types. Therefore, in the revised version of our methodology, we improved the L2A+ dust retrievals by incorporating CAMS reanalysis data. CAMS-derived aerosol information was used to estimate particle depolarization ratios dynamically through the one-step POLIPHON method, enabling a physically consistent separation of dust and non-dust components and the derivation of an improved Aeolus L2A+ dust product, analogous to the approach used in the CALIPSO LIVAS database. In the figure

below, we provide the CAMS-derived particle circular depolarization ratio profiles for the case study on 3$^{rd}$ September 2021 at the Aeolus horizontal (per BRC) and vertical scale (24 vertical bins).

[Figure]

Figure 1: a) Particle circular depolarization ratio vertical cross sections along the Aeolus orbit over Mindelo, Cabo Verde (orbit id: 017568) at the regular vertical scale of Aeolus.

The new methodology is now ready to be implemented on a global scale, allowing the production of an enhanced Aeolus dust product with improved accuracy and consistency across different aerosol environments.

Regarding ground-based depolarization measurements, particle linear depolarization ratio data were available only from the eVe lidar system operated during the ASKOS campaign, while the PollyXT system did not provide depolarization measurements for the examined period. However, in our analysis, the depolarization ratio information used to reconstruct the missing Aeolus cross-polar component was derived entirely from CAMS reanalysis outputs through the one-step POLIPHON methodology, rather than from Aeolus observations themselves. For this reason, we did not perform a direct comparison between the CAMS-derived depolarization ratios and the eVe ground-based measurements, since the CAMS-based ratios represent modeled, reanalysis-driven estimates and not Aeolus-measured quantities. A comparison with eVe depolarization data would therefore not serve as a direct validation of the Aeolus-based retrievals but rather of the CAMS product, which lies beyond the scope of this study. This type of comparison could be incorporated in a future study focused on validating CAMS-derived depolarization ratios against both satellite-based and ground-based lidar measurements, further supporting the reliability of CAMS data for aerosol typing applications.

5.  *The L2A products labelled Baseline 16 include Quality Checks (QC) flags for SCA, SCA-MB, and MLE (ATBD, Flamant et al., 2022, and User Guide, Trapon et al., 2022). These flags are of high relevance, especially for noisy signals derived with the SCA in low Signal-to-Noise regions and for MLE in low altitudes below 2 km. Could the authors clarify if these quality flags have*

*been applied to the signals ? If not applied, is it because bins removal and the will to keep as*
*many valid bins as possible before cloud screening? A few lines discussing this point can be*
*added to the text, at least saying that ready-to-use QC flags are included in the L2A product.*

**Response:**

We appreciate the reviewer's observation regarding the Quality Check (QC) flags included in the Aeolus
L2A Baseline 16 products. In the current study, the Aeolus QC flags were not applied to the SCA, SCA-
MB, or MLE retrievals. This decision was made to avoid a substantial reduction in the number of valid bins
per profile per BRC, which would have significantly limited the statistical robustness of the Aeolus–ground-
based comparisons. As suggested, we have now clarified this point in the methodology section of the
revised manuscript, explicitly noting that the QC flags were not used and explaining the rationale behind
this decision.

6. *The L2A products labelled Baseline 16 include newly implemented MLE processed at finer*
   *horizontal resolution sub-BRC (MLEsub) which was implemented in L2A processor version*
   *3.16. Each sub profile corresponds to 6 accumulated measurements with September 2021*
   *settings. A BRC is then filled with 5 MLEsub sub-profiles .If considering the comments 1), it*
   *would be interesting to reproduce the study looking at MLEsub versus AEL_PRO ; the averaging*
   *for AEL_PRO being then less prominent and possibly impacting the cloud screening. Kindly*
   *consider mentioning this open point into the conclusion section, or eventually within a code and*
   *data availability section.*

**Response:**

Indeed, the recent implementation of the MLEsub product in the Aeolus L2A Baseline 16 (processor
version 3.16) provides improved horizontal resolution by reducing the averaging scale to sub-BRC level,
corresponding to approximately 15–18 km per sub-profile. As demonstrated by Trapon et al. (2025), this
finer resolution significantly enhances the representation of aerosol layers and mitigates the impact of
averaging dilution, particularly in regions with strong spatial variability or partial cloud contamination.

Although our present study relies on SCA, SCA-MB, AEL_PRO and MLE datasets at the standard BRC
resolution, we recognize that the use of MLEsub could further improve the cloud-screening accuracy and
aerosol retrieval consistency. Following the reviewer's suggestion, we have added a statement in the
conclusion section noting this open point and highlighting that future work will include a comparative

analysis of Aeolus optical products at BRC scale (e.g., SCA, MLE) versus MLEsub, in order to assess the impact of horizontal averaging on the retrieval of dust and mixed aerosol layers.

*Specific comments:*

1. *Line 104 "...through the independent estimation of the the volume extinction and co-polarized volume backscatter coefficient...". This is because the ability to seperate the contributions from molecules and particles with the two channels. It can be mentionned here as well (it is mentionned later in the paper by section 2.1 line 188).*

**Response:**

We thank the reviewer for the suggestion.We corrected this point.

2. *Line 106 "…from two different spectral channels (Ehlers et al., 2022)…". Kindly consider (Flamant et al., 2008) instead.*

**Response:**

We corrected this point.

3. *Line 116 "…reprocessed in Baseline 10 (B10)…". Kindly consider rephrasing, i.e. L2A processor version 3.10, referred as Baseline 10 (B10) .*

**Response:**

We corrected this point.

4. *Line 124 "…underestimation of the order of 18% of the Aeolus-like backscatter…". What does Aeolus-like backscatter mean ? Do the authors refer to the underestimation of the total backscatter ? Kindly clarify.*

**Response:**

This is a study of Paschou et al. (2022). In this study, the Aeolus-like backscatter coefficient –measured by eVe lidar- -corresponded to the co-polar particle backscatter coefficient that Aeolus would measure from the ground. The results showed an underestimation of the order of ~ 18 % of the Aeolus like backscatter coefficient when moderately depolarizing dust particles are probed.

5. *Line 187 "…cross-talk coefficients…". The principle of the cross-talk correction, i.e. seperation between lidar signal contributions from particles and molecules for vertical matching between Rayleigh an Mie channels, can be mentioned here.*

**Response:**

This point has now been corrected as suggested by the reviewer, and the sentence has been revised to include an explanation of the role of the cross-talk coefficients.

6. *Line 188 "…and molecular and particulate contributions to the signals…". The reference papers (Flament et al., 2020 ; Flament et al., 2021) can be added.*

**Response:**

We thank the reviewer for the suggestion. The references to Flament et al. (2020) and Flament et al. (2021) have now been added to the sentence to support the description of the molecular and particulate contributions to the signals.

7. *Line 507 "…To address this issue…". Would limitation be more suited than issue ?*

**Response:**

We thank the reviewer for the suggestion. This point has been corrected in the revised version of the manuscript.

8. *Figures 9-10-11 If considering high dust fraction up to ~ 70 % between ~ 10 km altitude and ~ 16 km altitude between profiles [-21.87°E, 0.38°N] and [-25.28°E, 18.62°N] on case 3rd September 2021 in Fig. 9b, why do we see only white background for the corresponding regions in SCA and MLE retrievals in Figs 10-11 ? Does that mean fully cloud contaminated bins which were then removed, or invalid SCA and MLE ? But SCA and MLE products do not seem too much attenuated below this regions, then between ground and ~ 10 km...The flagging method including dust-free removal and cloudy contaminated bins is difficult to apprehend for such regions. Are there any other quality flagging applied here ? Is it because unrealistic dust fraction by CAMS ? Kindly clarify.*

**Response:**

We thank the reviewer for this insightful comment. The regions appearing as white background in the SCA and MLE retrievals (Figs. 10–11) correspond to bins that were identified as cloud-contaminated. Specifically, these bins were flagged as cloudy through the implementation of the synergistic AEL-FM and SEVIRI cloud-mask product. Following our quality-control procedure, all cloud-affected bins are removed from the SCA and MLE analyses to ensure the reliability of the retrieved aerosol properties. Therefore, the absence of retrievals in those altitude ranges is not related to unrealistic dust fractions from CAMS, nor to attenuation issues in the SCA or MLE products; it simply reflects the exclusion of bins classified as cloud-contaminated.

9. *Figure 11 Kindly consider changing the colormap to be able to visualy compare the low dust mass concentration (i.e. below 150 μg/m³) from the suroundings.*

**Response:**

We thank the reviewer for this helpful suggestion. We agree that the current colormap in Figure 11 does not provide sufficient visual contrast for low dust mass concentration values (below 150 μg m⁻³). In the revised version of the manuscript, we updated the colormap to enhance the visibility and distinction of the lower concentration ranges, thereby improving the interpretability of the figure.

10. *Figure 11 The DEM intersection is visible for pannels Fig. 11a-c-e-g but not for pannels Fig. 11b-d-f-h. This may be linked to a linewidth adjustment. Kindly consider re-generation of the figures.*

**Response:**

We thank the reviewer for pointing out the inconsistency regarding the visibility of the DEM intersection across the panels in Figure 11. The issue is indeed related to a linewidth inconsistency during figure generation. We will regenerate all panels of Figure 11 using a uniform linewidth setting to ensure the DEM intersection is clearly and consistently visible in all subfigures (11a–h) in the revised manuscript.

11. *Figures 12-13 The vertical profiles and errors in dashed lines are hardly dinstinguishable. Kindly consider 2x2 pannels with increased size instead, and possibly reducing the xaxis top limit of backscatter coefficient. Moreover, kindly consider other color combination than red-green to get colorblind-safe color scheme.*

**Response:**

We thank the reviewer for this constructive comment. To improve clarity, we have reduced the alpha (transparency) of the dashed lines representing errors in the Python script, making the profiles and errors more distinguishable while avoiding confusion. Additionally, we increased the linewidth of the vertical profiles to enhance visibility.

12. *Figures 14-15 Kindly consider adjustment of yaxis and xaxis to lower values (e.g. 6 or 8 Mm⁻¹sr⁻¹) for better readability.*

**Response:**

We thank the reviewer for the valuable suggestion. In the revised manuscript, the number of scatterplot points has been reduced by including only the dust-affected bin layers. Combined with the adjusted x- and y-axis limits (0-10 Mm⁻¹sr⁻¹) , this reduction facilitates the readability of the figures and allows the data trends to be more clearly observed.